# WHAT IS PREFERENCE OPTIMIZATION DOING, HOW AND WHY?

## ABSTRACT

*Preference optimization* (PO) is indispensable for *large language models* (LLMs), with methods such as *direct preference optimization* (DPO) and *proximal policy optimization* (PPO) achieving great success. A common belief is that DPO is supervised learning while PPO is reinforcement learning, yet deeper analyses for the reasons underlying these differences remain lacking. To fill this gap, we analyze their *optimization dynamics*, revealing distinct algorithmic behaviors and comprehending their underlying causes. First, we examine the target directions of gradient-based updates and find that DPO follows stable targets, whereas PPO follows dynamic targets that balance exploration and exploitation, thus validating the common belief from a new perspective. Second, we examine the roles of positive learning, negative learning, and loss reweighting, which are three key components in PO methods. Our analyses reveal that these components play fairly different roles. In DPO, positive and negative learning jointly shape the learning targets meanwhile mutually offset each other. However, loss reweighting in DPO acts less as a reward signal but more as a regularizer to mitigate overfitting. In PPO, negative learning primarily supports exploration rather than determining the targets. Meanwhile, loss reweighting, related to absolute values of token-level advantages, indicates the distinct roles of token groups in updating targets. Given these findings, we conduct carefully designed ablation studies to further examine how controlling these dynamics impacts optimization efficiency and practical performance. The insights gained from our analyses not only deepen the understanding of PO methods but also inspire the development of more preference-aligned LLMs.

## 1 INTRODUCTION

*Preference optimization* (PO) (Sutton et al., 1998) plays an essential role in fine-tuning *large language models* (LLMs), enabling alignment with human preferences (Achiam et al., 2023) and tackling complex reasoning tasks (Liu et al., 2024a). Its principle of maximizing rewards serves as the foundation for policy gradient and its advanced alternatives, such as *proximal policy optimization* (PPO) (Schulman et al., 2017) and *direct preference optimization* (DPO) (Rafailov et al., 2024), continuing to benefit research and business. These methods follow distinct learning paradigms and can be further viewed as incorporating diverse operations, such as positive learning, negative learning, and loss reweighting, cf., Section 3, whose combinations often lead to strong and well-deserved performance.

To further understand the current success of DPO and PPO and advance beyond them, it is essential to analyze the causes of their differences and to quantify the individual behaviors of these components, which is a kind of study that has been seldom conducted. This becomes particularly important in light of conflicting findings in related studies, such as evidences suggesting that negative learning can harm generalization (Wang et al., 2025) and that loss reweighting has limited impact on over-parameterized models (Zhai et al., 2022; Shao et al., 2025). Thorough analyses are critical for comprehending the behaviors of current PO methods and for inspiring new strategies to drive further improvements. However, achieving this ultimate goal is not easy, as we even lack proper tools to understand the overall PO behaviors—a gap that should be addressed first. Learning behaviors broadly fall into one of two paradigms: supervised learning and reinforcement learning. To be precise, *supervised learning* is characterized by relatively clear and stable targets, reflected explicitly from its gradient directions. In contrast, *reinforcement learning* has more indirect and dynamic learning targets. While the reward function is known, there is no precise gradient signal to maximize it, necessitating exploration.

In this paper, we identify the learning behaviors by showing the stability of their optimization dynamics in achieving the final responses, going beyond conventional judgments based solely on objective forms, since the same objective can reflect either supervised or reinforcement learning depending on data distributions and base models. The stability is measured through the gradient dot products between the learning objectives to be minimized and the expected negative log-likelihood on the final responses, with respect to a sequence of PO checkpoints, cf., equation 3. Overall, positive dot products throughout learning indicate stable, targeted learning, while near-zero, negative, or fluctuating values reflect exploratory behaviors. We conducted rigorous experiments on two representative PO methods for the task of human preference alignment: DPO, which contrastively learns from predefined win-lose pairs in an offline manner, and PPO, an online method that rolls out candidate tokens and updates the model based on their associated advantages. The detailed analyses are presented in Section 3, and we summarize the key observations as follows.

- *DPO aligns with supervised learning behavior*. However, its learning targets are implicit, relying not only on win data but also on lose data, and potentially on the model parameters.
- *PPO follows reinforcement learning behavior*. Its exploration is stable, mainly along orthogonal targets, yet with a slight shift toward conflicting candidates, showing its broad coverage.

These observations align well with the common belief, and this part of the analyses serves as a further verification, albeit from a new perspective of optimization dynamics. The true value of our analysis framework lies in uncovering the mechanisms underlying these observations, enabling us to disentangle the interactions of additive components by examining their respective gradients. Detailed analyses are also presented in Section 3, with the key observations as follows.

- *Positive and negative learning*, maximizing and minimizing likelihood respectively, play distinct roles. In DPO, they jointly shape the learning targets, with the influence of negative learning initially weak but gradually dominating that of positive learning. In PPO, positive learning alone contributes to shaping targets while negative learning primarily ensures exploration.
- *Loss reweighting*, reflected in implicit rewards in DPO and absolute advantages in PPO, also plays distinct roles across methods. In DPO, it downweights fully learned data without offering useful reward signals, whereas in PPO, loss reweighting conveys meaningful information more than raw advantages to indicate the learning dynamics.

Our analyses above focus on the learning dynamics involved in *achieving final responses*, identifying the objective- and component-level effects within DPO and PPO. While offering valuable insights, *achieving improved performance* remains one of our key goals to contribute to the community, please refer to Appendix A for a discussion of their subtle difference. To bridge the gap, we pose the question of how controlling the learning dynamics impacts model performance. We test several promising ways of behavior control, adjusting the influence of individual components within DPO and PPO. Ablation studies are presented in Section 4, with main conclusions as follows.

- Components that hinder the achievement of final responses may benefit the performance, thus *acting as regularization*. Carefully adjusting their influence can yield further improvements.
- PO methods with reinforcement behavior can benefit from *slight shifts toward supervised learning*. Yet, excessive shifting would risk rapid overfitting to current responses.

To conclude, in Section 5, we situate our findings within the broader landscape of well-tested tricks and latest methods, interpreting these practices through our new aspect of optimization dynamics. We further envision promising directions that may open pathways toward new learning paradigms, with preliminary experiments in Appendix D demonstrating their potential.

## 2 CONCEPTS: LEARNING BEHAVIORS AND OPTIMIZATION DYNAMICS

To begin, we discuss our ways for analyzing PO behaviors rigorously. As aforementioned, we attribute learning behaviors into two categories: supervised learning and reinforcement learning, representing two primary paradigms (Sutton et al., 1998; Mohri et al., 2018). Their essential difference lies in whether they learn from demonstrations or rewards. However, this distinction has been widely discussed and may yield limited additional insight. Instead, we adopt an alternative viewpoint and compare them in terms of whether they possess relatively stable training targets, as detailed below.

- *Supervised learning* is characterized by stable targets reflected in its learning objectives explicitly or implicitly. It provides clear gradient signals, where optimization dynamics are expected to steadily progress toward the corresponding targets. For example, for the question *What is the capital of France?*, the exact answer *Paris* serves as a clear learning target. Ideally, the model can be supervised to maximize the log-likelihood of tokens related to *Paris*.
- *Reinforcement learning* involves exploration and exploitation jointly. Its optimization procedure searches for improved targets, indicated by increased rewards, to establish new gradient signals to learn. For the same question, the model may initially roll out incorrect responses, such as *Tokyo*. The reward function will judge these answers as incorrect, prompting the model to continue exploring until identifying the correct one, i.e., *Paris*.

Analyzing learning behaviors enables deeper insight into PO methods. As shown later, this not only sheds light on interpreting existing methods but also highlights pathways for further improvements.

## 2.1 ANALYSIS TOOL

We observe that targets are key to distinguishing between supervised and reinforcement learning: the former relies on stable targets, whereas the latter involves fluctuating ones. However, we cannot directly access such targets, as they are typically implicit, and it is hard to measure their similarity, as the measurement is typically model-dependent (Koh & Liang, 2017; Ren & Sutherland, 2024). An alternative is to examine whether the procedure for obtaining the final responses is stable, assuming sufficient training duration and evaluation questions similar to those for training. Specifically, if learning targets are stable, most training steps would contribute to the final responses. On the contrary, if the targets are unstable, intermediate steps may point towards orthogonal or even conflicting targets compared to the later phase, failing to benefit the final responses. This insight motivates an analysis framework as follows to characterize the learning behaviors through their optimization dynamics.

**Notations.** Consider an LLM that defines the conditional distribution $\pi(y|x; \boldsymbol{\theta})$ with $x$ the question, $y$ the response, and $\boldsymbol{\theta}$ the learnable parameters. In general, the PO problem is formulated as $\max_{\boldsymbol{\theta}} \left[ \mathcal{J}(\pi; \boldsymbol{\theta}) := \mathbb{E}_{\pi_{\boldsymbol{\theta}}} R(x, y) \right]$, where $\mathbb{E}_{\pi_{\boldsymbol{\theta}}}$ denotes the expectation over the conditional distribution $\pi(y|x; \boldsymbol{\theta})$ and $R$ represents the reward model for the task of interest. However, this problem is inherently non-differentiable, as it depends on sampled outcomes rather than directly on differentiable quantities such as likelihoods, and is therefore intractable for optimization. Therefore, researchers turn to explore its alternatives. Specifically, consider the training dataset $\mathcal{D} = \{(x, y)\}_n$ of size $n$, which contains question-response data. Here, $y$ may either be predefined or sampled, corresponding to offline and online settings, respectively. The surrogate objective $\mathcal{L}$ is formulated as $\min_{\boldsymbol{\theta}} \left[ \mathcal{L}(\mathcal{D}; \boldsymbol{\theta}) := \hat{\mathbb{E}}_{\mathcal{D}} \ell(x, y; \boldsymbol{\theta}) \right]$, where $\hat{\mathbb{E}}_{\mathcal{D}}$ is the empirical expectation computed over the training dataset $\mathcal{D}$ and $\ell$ is a differentiable loss function that approximates the process of maximizing $\mathcal{J}$. If necessary, we use $\boldsymbol{\theta}_{(t)}$ to denote the parameters at step $t$ and $\boldsymbol{\theta}^{\mathrm{po}}$ for that after PO training.

**A Formal Framework.** We introduce the final responses dataset $\mathcal{D}' = \{(x', y')\}_{n'}$ of size $n'$, comprising outputs from $\pi(y|x; \boldsymbol{\theta}^{\mathrm{po}})$ based on the greedy search, with $x'$ related to $\mathcal{D}$ but not necessarily included in it, allowing us to consider out-of-distribution generalization (Ye et al., 2021). Then, we say the learning objective $\mathcal{L}$ at step $t$ can benefit the optimization dynamics in achieving the final responses when it can reduce the expected negative log-likelihood over $\mathcal{D}'$, which is

$$\hat{\mathbb{E}}_{\mathcal{D}'} \left[ -\log \pi(y'|x'; \boldsymbol{\theta}_{(t+1)}) \right] < \hat{\mathbb{E}}_{\mathcal{D}'} \left[ -\log \pi(y'|x'; \boldsymbol{\theta}_{(t)}) \right], \tag{1}$$

where $\boldsymbol{\theta}_{(t+1)} = \boldsymbol{\theta}_{(t)} - \eta \nabla_{\boldsymbol{\theta}} \mathcal{L}(\mathcal{D}; \boldsymbol{\theta}_{(t)})$ is the parameter update at the step $t$ with the learning rate $\eta$. However, the log-likelihood alone offers limited insight, especially for our component-level analyses later in Section 3. This motivates the Taylor expansion of equation 1, expanding the left-hand side to the first order around $\boldsymbol{\theta}_{(t)}$, yielding

$$\begin{aligned} &\hat{\mathbb{E}}_{\mathcal{D}'} \left[ -\log \pi(y'|x'; \boldsymbol{\theta}_{(t+1)}) \right] \\ =&\hat{\mathbb{E}}_{\mathcal{D}'} \left[ -\log \pi(y'|x'; \boldsymbol{\theta}_{(t)}) + \eta \nabla_{\boldsymbol{\theta}} \log \pi(y'|x'; \boldsymbol{\theta}_{(t)})^\top \nabla_{\boldsymbol{\theta}} \mathcal{L}(\mathcal{D}; \boldsymbol{\theta}_{(t)}) + \mathcal{O}(\eta^2) \right]. \end{aligned} \tag{2}$$

Substituting it back and neglecting high-order terms, we have the *gradient alignment* condition of

$$\mathcal{G}(\mathcal{L}; \boldsymbol{\theta}_{(t)}) = \hat{\mathbb{E}}_{\mathcal{D}'} \left[ -\nabla_{\boldsymbol{\theta}} \log \pi(y'|x'; \boldsymbol{\theta}_{(t)}) \right]^\top \nabla_{\boldsymbol{\theta}} \mathcal{L}(\mathcal{D}; \boldsymbol{\theta}_{(t)}) > 0, \tag{3}$$

indicating that $\mathcal{L}$ improves the likelihood on $\mathcal{D}'$ at the step $t$. Thus, we regard the learning dynamic as supervised when $\mathcal{G}(\mathcal{L}; \boldsymbol{\theta}_{(t)})$ keeps positive for the majority of $t$. Moreover, when $\mathcal{G}(\mathcal{L}; \boldsymbol{\theta}_{(t)})$ is close

to zero, negative, or fluctuates between positive and negative, it is reinforcement learning. Please refer to Appendix A for a detailed discussion of the limitations of these approximations and several additional important remarks, and to Appendix E.1 for more formal and rigorous derivations.

# 3 KEY ANALYSES: DPO AND PPO AS CASE STUDIES

We are now able to conduct formal analyses to understand the exact learning behaviors exhibited by different PO methods. We focus on the task of preference alignment, examining two PO methods: DPO and PPO, which are the most important offline and online methods, respectively. As case studies, our experiments are based on the pre-trained Pythia-2.8B base model (Biderman et al., 2023), with the UltraChat-200k dataset (Ding et al., 2023) for *supervised fine-tuning* (SFT), the UltraFeedback dataset (Cui et al., 2024) for PO training, and the HH-RLHF-helpfulness dataset (Bai et al., 2022) to construct the final responses dataset. Note that only the question part of HH-RLHF-helpfulness was used, while the answer part was generated via the greedy search on the PO-trained models. Please refer to Appendix B for additional details, and to Appendices C and D for results on other models, such as Qwen3-1.7B (Yang et al., 2025) and Llama3-8B (Dubey et al., 2024).

## 3.1 DPO

We begin our analyses of DPO by first reviewing its key derivations and disentangling the core components. We will then examine the overall optimization dynamics as well as component-wise effects, with our primary goals of understanding its learning behaviors and gaining some new insights.

**A Brief Review.** Further with the Kullback-Leibler constraint (Peters & Schaal, 2007), the optimal policy with parameters $\boldsymbol{\theta}^*$, derived from $\mathcal{J}$, admits a closed-form solution satisfying $R(x, y) = \beta \log \frac{\pi(y|x;\boldsymbol{\theta}^*)}{\pi(y|x;\boldsymbol{\theta}^{\mathrm{ref}})} + \beta \log Z(x)$, where $\boldsymbol{\theta}^{\mathrm{ref}}$ denotes the parameters of a reference model, $\beta$ is the hyper-parameter, and $Z(\cdot)$ is the partition function. Assuming the Bradley-Terry modeling (Hunter, 2004), the preference probability can then be derived as $p(y^+ \succ y^-|x) = \sigma\big[\beta \log \frac{\pi(y^+|x;\boldsymbol{\theta}^*)}{\pi(y^+|x;\boldsymbol{\theta}^{\mathrm{ref}})} - \beta \log \frac{\pi(y^-|x;\boldsymbol{\theta}^*)}{\pi(y^-|x;\boldsymbol{\theta}^{\mathrm{ref}})}\big]$ with $\sigma(\cdot)$ the sigmoid function. DPO leverages this preference model to construct the learning objective, minimizing the negative log-likelihood of the observed preferences, as follows

$$\mathcal{L}_{\mathrm{dpo}}(\boldsymbol{\theta}) = -\hat{\mathbb{E}}_{\mathcal{D}}\left[\log \sigma\Big(\beta\big(\log \frac{\pi(y^+|x;\boldsymbol{\theta})}{\pi(y^+|x;\boldsymbol{\theta}^{\mathrm{ref}})} - \log \frac{\pi(y^-|x;\boldsymbol{\theta})}{\pi(y^-|x;\boldsymbol{\theta}^{\mathrm{ref}})}\big)\Big)\right], \quad (4)$$

where $\mathcal{D} = \{(x, y^+, y^-)\}_n$ is the dataset of preference pairs, with $y^+$ being preferred over $y^-$.

**Key Components.** Equation 4 is complex. To make it clear, we present a gradient-equivalent form as

$$\hat{\mathcal{L}}_{\mathrm{dpo}}(\boldsymbol{\theta}) = -\hat{\mathbb{E}}_{\mathcal{D}}\left[\omega(x, y^+, y^-)\big(\log \pi(y^+|x;\boldsymbol{\theta}) - \log \pi(y^-|x;\boldsymbol{\theta})\big)\right], \quad (5)$$

where $\omega(x, y^+, y^-) = \beta\sigma\big[-\beta \log \frac{\pi(y^+|x;\boldsymbol{\theta})}{\pi(y^-|x;\boldsymbol{\theta})} + \beta \log \frac{\pi(y^+|x;\boldsymbol{\theta}^{\mathrm{ref}})}{\pi(y^-|x;\boldsymbol{\theta}^{\mathrm{ref}})}\big]$ is the implicit reward function (Rafailov et al., 2024). It becomes clear that DPO aims to increase the likelihood of $y^+$ while decreasing that of $y^-$. This procedure is further reweighted by $\omega$. It indicates that the learning objective of DPO comprises three components: positive learning $\log \pi(y^+|x;\boldsymbol{\theta})$, which increases likelihood; negative learning $-\log \pi(y^-|x;\boldsymbol{\theta})$, which decreases likelihood; and loss reweighting $\omega(x, y^+, y^-)$, which adjusts the importance of data during training. We can identify their respective roles during training. When analyzing the impacts of positive and negative learning, we have

$$\mathcal{L}_{\mathrm{dpo}}^+(\boldsymbol{\theta}) = \hat{\mathbb{E}}_{\mathcal{D}}\big[-\omega(x, y^+, y^-)\log \pi(y^+|x;\boldsymbol{\theta})\big], \quad (6)$$

$$\mathcal{L}_{\mathrm{dpo}}^-(\boldsymbol{\theta}) = \hat{\mathbb{E}}_{\mathcal{D}}\big[\omega(x, y^+, y^-)\log \pi(y^-|x;\boldsymbol{\theta}))\big]. \quad (7)$$

Therein, $\mathcal{L}_{\mathrm{dpo}}^+$ and $\mathcal{L}_{\mathrm{dpo}}^-$ represent the components of positive and negative learning, respectively. Moreover, when analyzing the impacts of loss reweighting, samples are divided into three groups

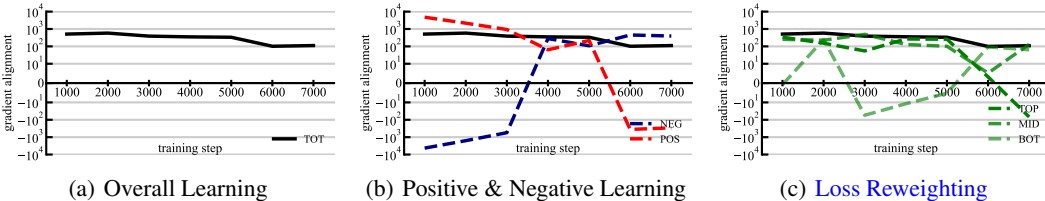

(a) Overall Learning        (b) Positive & Negative Learning        (c) Loss Reweighting

Figure 1: *DPO Learning Dynamics*. For the Pythia-2.8B model trained on UltraChat 200K and tested on HH-RLHF-helpfulness, we show the dynamics of $\mathcal{G}$ measured per 1000 training steps: (a) the overall objective $\mathcal{L}_{\text{dpo}}$ (TOT); (b) the positive $\mathcal{L}_{\text{dpo}}^+$ (POS) and negative $\mathcal{L}_{\text{dpo}}^-$ (NEG) components; and (c) the weighted top $\mathcal{L}_{\text{dpo}}^\uparrow$ (TOP), middle $\mathcal{L}_{\text{dpo}}^\rightarrow$ (MID), and bottom $\mathcal{L}_{\text{dpo}}^\downarrow$ (BOT) components. The log scale is used for $\mathcal{G}$ due to its span across several orders of magnitude.

based on their ranking of $\omega$, which is given by

$$\mathcal{L}_{\text{dpo}}^\uparrow(\boldsymbol{\theta}) = -\hat{\mathbb{E}}_{\mathcal{D}}\left[\omega(x, y^+, y^-)\big(\log \pi(y^+|x; \boldsymbol{\theta}) - \log \pi(y^-|x; \boldsymbol{\theta})\big)\mathbb{1}_{\{\omega(x,y^+,y^-) \geq Q_{2/3}(\omega)\}}\right], \quad (8)$$

$$\mathcal{L}_{\text{dpo}}^\rightarrow(\boldsymbol{\theta}) = -\hat{\mathbb{E}}_{\mathcal{D}}\left[\omega(x, y^+, y^-)\big(\log \pi(y^+|x; \boldsymbol{\theta}) - \log \pi(y^-|x; \boldsymbol{\theta})\big)\mathbb{1}_{\{Q_{1/3}(\omega) < \omega(x,y^+,y^-) < Q_{2/3}(\omega)\}}\right], \quad (9)$$

$$\mathcal{L}_{\text{dpo}}^\downarrow(\boldsymbol{\theta}) = -\hat{\mathbb{E}}_{\mathcal{D}}\left[\omega(x, y^+, y^-)\big(\log \pi(y^+|x; \boldsymbol{\theta}) - \log \pi(y^-|x; \boldsymbol{\theta})\big)\mathbb{1}_{\{\omega(x,y^+,y^-) \leq Q_{1/3}(\omega)\}}\right]. \quad (10)$$

$\mathcal{L}_{\text{dpo}}^\uparrow$, $\mathcal{L}_{\text{dpo}}^\rightarrow$, and $\mathcal{L}_{\text{dpo}}^\downarrow$ represent the top, middle, and bottom weighted parts, respectively. Therein, $\mathbb{1}$ denotes the indicator function and $Q_\alpha(\omega)$ refers to the $\alpha$-quantile for the distribution of $\omega$.

**Learning Behaviors.** Now, we examine the *overall behaviors*, reporting $\mathcal{G}(\mathcal{L}_{\text{dpo}}; \boldsymbol{\theta}_{(t)})$ per 1000 steps. In Figure 1(a), we observe that $\mathcal{G}$ keeps positive and notably larger than 0, indicating stable targets with supervised behaviors. This observation is consistent with the prevailing view, yet specific components that contribute to shaping these learning targets remain unclear and merit further investigation. In Figure 1(b), we monitored $\mathcal{G}(\mathcal{L}_{\text{dpo}}^+; \boldsymbol{\theta}_{(t)})$ for *positive learning* and $\mathcal{G}(\mathcal{L}_{\text{dpo}}^-; \boldsymbol{\theta}_{(t)})$ for *negative learning*. At the start, positive learning takes the lead in shaping the targets, while negative learning acts to offset its impacts, preventing an excessive reliance on $y^+$. This interaction enables the actual targets to diverge from $y^+$. As training progresses, a surprising aspect is that negative learning begins to shape the learning targets around step 3500. Following this, positive learning gradually shifts roles, countering the effects of negative learning after about step 5500.

A plausible explanation is that the gradient magnitude of positive learning initially dominates but gradually diminishes, while that of negative learning, moving in the opposite direction, amplifies and eventually dominates. However, as shown in Appendix C, this assumption is incorrect. Instead, we found that the gradient magnitudes of positive and negative learning remain of the same order throughout DPO training, cf., Figure 8(a), suggesting that the overall gradient direction benefits roughly from both. Therefore, we propose the following assumption: the explicit targets associated with positive learning, i.e., $y^+$, contribute to the final responses in the early stages but eventually lead to overfitting, causing $\mathcal{G}$ to become negative later. In contrast, as training progresses and the model becomes stronger, negative learning can provide implicit, model-dependent target directions, squeezing the likelihood regions of $y^-$ amplifies other high-likelihood regions (Ren & Sutherland, 2025). We illustrate in Appendix C that this assumption of overfitting is reasonable.

Moreover, in Figure 1(c), we illustrate the influence of the top, middle, and bottom *loss reweighting* via $\mathcal{G}(\mathcal{L}_{\text{dpo}}^\uparrow; \boldsymbol{\theta}_{(t)})$, $\mathcal{G}(\mathcal{L}_{\text{dpo}}^\rightarrow; \boldsymbol{\theta}_{(t)})$, and $\mathcal{G}(\mathcal{L}_{\text{dpo}}^\downarrow; \boldsymbol{\theta}_{(t)})$, respectively. Although not entirely unexpected, we observed that $\omega$ does not act as a reliable reward function, as higher weighted data do not consistently contribute more to the learning targets. This arises because the unbiasedness of the surrogate objective $\mathcal{L}_{\text{dpo}}$ regarding the true objective $\mathcal{J}$ holds only at the optimal policy with parameters $\boldsymbol{\theta}^*$, which is unavailable in advance, presenting a chicken-and-egg dilemma. Instead, from a classical viewpoint of loss reweighting (Sugiyama & Kawanabe, 2012; Lodkaew et al., 2025), we can interpret the role of $\omega$ in a more straightforward way: it places greater emphasis on data with insufficiently large margins of likelihood between $y^+$ and $y^-$, a common mechanism to tackle overfitting.

**Summary.** We are now ready to answer the question: "What is DPO doing, how and why?" The mutual impacts of positive and negative learning jointly shape the implicit targets. Together, they drive a supervised learning dynamics, aligning well with our intuition, as DPO operates on a fixed dataset and learns from it contrastingly. Upon closer examination, we know that $y^+$ and $y^-$ interact in a mutually constraining manner most of the time. In the early phase, positive learning shapes the targets towards $y^+$, while negative learning prevents the model from overly relying on $y^+$. In the later phase, negative learning gradually takes the lead in shaping the targets. It does so indirectly by reducing the probability mass associated with $y^-$ and amplifying that allocated to other high-likelihood regions, an assumption borrowed from (Ren & Sutherland, 2025). Moreover, positive learning remains crucial, yet taking on the role of preventing overly negative learning from collapsing the model (Wang et al., 2025). The implicit rewarding, which is a key claim in the original DPO paper, is not reliable: $\omega$ is dynamic and model-dependent, and the shaped targets are not predominantly influenced by the high-weighted parts, which are evidence for its deficiency as a reliable and effective reward mechanism. We propose that the primary role of $\omega$ is to act as a regularizer that mitigates overfitting: for well-learned data pairs with sufficiently small preference margin, i.e., small $\log \pi(y^+|x; \boldsymbol{\theta}) - \log \pi(y^-|x; \boldsymbol{\theta})$, DPO will assign the corresponding small $\omega$, thereby reducing its focus on these samples. This behavior will mitigate overfitting and thus functions as a form of regularization.

### 3.2 PPO

Now, we turn to PPO. As with DPO, we begin by reviewing its key derivations and core components. We will then analyze its learning behaviors, aiming to gain insights into the underlying mechanisms that contribute to its success.

**A Brief Review.** The likelihood-ratio trick (Williams, 1992) shows that $\mathcal{J}$ is differentiable, yielding the policy gradient $\nabla_{\boldsymbol{\theta}} \mathcal{J}(\boldsymbol{\theta}) = \mathbb{E}_{\pi_{\boldsymbol{\theta}}} [\nabla_{\boldsymbol{\theta}} \log \pi(y|x; \boldsymbol{\theta}) R(x, y)]$ or its token-wise expansion $\nabla_{\boldsymbol{\theta}} \mathcal{J}(\boldsymbol{\theta}) = \mathbb{E}_{\pi_{\boldsymbol{\theta}}} [\nabla_{\boldsymbol{\theta}} \log \pi(y|x; \boldsymbol{\theta}) A(x, y)]$, where $\pi_{\boldsymbol{\theta}}$ in the latter is reused to represent sampling for the next tokens, and $A(x, y)$ is the advantage function derived from $R$, which estimates the quality of the sampled tokens. PPO further enhances sampling efficiency by reusing data from the old model $\pi(y|x; \boldsymbol{\theta}_{\text{old}})$. Then, with importance sampling (Schulman et al., 2017), the policy gradient $\nabla_{\boldsymbol{\theta}} \mathcal{J}(\boldsymbol{\theta})$ can be approximated as $\mathbb{E}_{\pi_{\boldsymbol{\theta}_{\text{old}}}} [\frac{\pi(y|x; \boldsymbol{\theta})}{\pi(y|x; \boldsymbol{\theta}_{\text{old}})} \nabla_{\boldsymbol{\theta}} \log \pi(y|x; \boldsymbol{\theta}) \hat{A}(x, y)]$, where $\hat{A}(x, y)$ estimates $A(x, y)$ by $\pi_{\boldsymbol{\theta}_{\text{old}}}$ and is typically normalized. Removing the gradient operation, we derive the basic learning objective of PPO as $\mathcal{J}(\boldsymbol{\theta}) = \mathbb{E}_{\pi_{\boldsymbol{\theta}_{\text{old}}}} [\frac{\pi(y|x; \boldsymbol{\theta})}{\pi(y|x; \boldsymbol{\theta}_{\text{old}})} \hat{A}(x, y)]$. Nevertheless, the importance ratio $\frac{\pi(y|x; \boldsymbol{\theta})}{\pi(y|x; \boldsymbol{\theta}_{\text{old}})}$ can grow arbitrarily large as the two model diverge, leading to unstable and potentially harmful updates. In practice, PPO updates the old model frequently and applies a clipping operation to restrict the ratio values. Rewriting sampling from $\pi_{\boldsymbol{\theta}_{\text{old}}}$ as $\mathcal{D}$ to align $\mathcal{L}$, the PPO objective becomes

$$\mathcal{L}_{\text{ppo}}(\boldsymbol{\theta}) = -\hat{\mathbb{E}}_{\mathcal{D}} \left[ \text{CLIP}_{\hat{A}(x,y)} \left[ \frac{\pi(y|x; \boldsymbol{\theta})}{\pi(y|x; \boldsymbol{\theta}_{\text{old}})} \right] \hat{A}(x, y) \right], \tag{11}$$

where $\text{CLIP}_a(x) = \min(x, 1 + \epsilon)$ if $a \geq 0$ and $\text{CLIP}_a(x) = \max(x, 1 - \epsilon)$ otherwise, with $\epsilon$ the hyper-parameter. It sets upper and lower bounds based on whether $\hat{A}$ is positive or negative.

**Key Components.** PPO differs from DPO in many aspects. For example, PPO is typically applied in a token-wise, online manner with a pre-defined reward function, whereas DPO is applied sentence-wise, offline, with an implicit reward function. However, they also share notable similarities. First, the importance ratio $\frac{\pi(y|x; \boldsymbol{\theta})}{\pi(y|x; \boldsymbol{\theta}_{\text{old}})}$ in PPO can be viewed as an approximation of $\log \pi(y|x; \boldsymbol{\theta})$ in DPO, due to their similar gradients when $\boldsymbol{\theta}$ is close to $\boldsymbol{\theta}_{\text{old}}$. Second, the value of $\hat{A}$ in practice can inherently be negative, resembling the roles of positive and negative learning in DPO. These facts motivate us to separate the key components of PPO in the same way as for DPO.

When analyzing the impacts of positive and negative learning, equation 11 can be split into two separate parts that account for positive and negative learning, respectively. This can be formulated as

$$\mathcal{L}_{\text{ppo}}^+(\boldsymbol{\theta}) = -\hat{\mathbb{E}}_{\mathcal{D}} \left[ \text{CLIP}_{\hat{A}(x,y)} \left[ \frac{\pi(y|x; \boldsymbol{\theta})}{\pi(y|x; \boldsymbol{\theta}_{\text{old}})} \right] \hat{A}(x, y) \mathbb{1}_{\{\hat{A}(x,y) \geq 0\}} \right], \tag{12}$$

$$\mathcal{L}_{\text{ppo}}^-(\boldsymbol{\theta}) = -\hat{\mathbb{E}}_{\mathcal{D}} \left[ \text{CLIP}_{\hat{A}(x,y)} \left[ \frac{\pi(y|x; \boldsymbol{\theta})}{\pi(y|x; \boldsymbol{\theta}_{\text{old}})} \right] \hat{A}(x, y) \mathbb{1}_{\{\hat{A}(x,y) < 0\}} \right]. \tag{13}$$

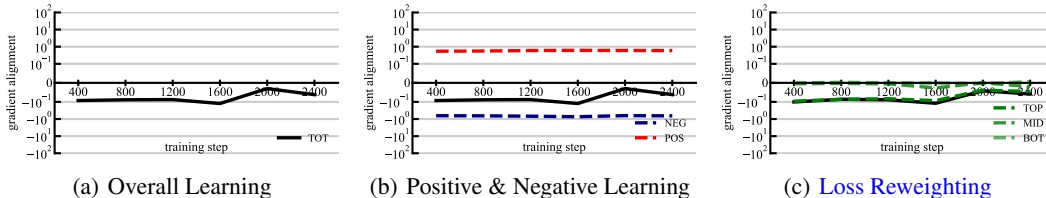

(a) Overall Learning     (b) Positive & Negative Learning     (c) Loss Reweighting

Figure 2: *PPO Learning Dynamics*. For the Pythia-2.8B model trained on UltraChat 200K and tested on HH-RLHF-helpfulness, we show the dynamics of $\mathcal{G}$ measured per $400$ training steps: (a) the overall objective $\mathcal{L}_{\mathrm{ppo}}$ (TOT); (b) the positive $\mathcal{L}_{\mathrm{ppo}}^+$ (POS) and negative $\mathcal{L}_{\mathrm{ppo}}^-$ (NEG) components; and (c) the weighted top $\mathcal{L}_{\mathrm{ppo}}^\uparrow$ (TOP), middle $\mathcal{L}_{\mathrm{ppo}}^\rightarrow$ (MID), and bottom $\mathcal{L}_{\mathrm{ppo}}^\downarrow$ (BOT) components.

For loss reweighting, data are divided into three levels based on their ranking of $|\hat{A}|$, namely,

$$\mathcal{L}_{\mathrm{ppo}}^\uparrow(\boldsymbol{\theta}) = -\hat{\mathbb{E}}_\mathcal{D}\left[\mathtt{CLIP}_{\hat{A}(x,y)}\left[\frac{\pi(y|x;\boldsymbol{\theta})}{\pi(y|x;\boldsymbol{\theta}_{\mathrm{old}})}\right]\hat{A}(x,y)\mathbb{1}_{\{|\hat{A}(x,y)|\geq Q_{2/3}(|\hat{A}|)\}}\right], \tag{14}$$

$$\mathcal{L}_{\mathrm{ppo}}^\rightarrow(\boldsymbol{\theta}) = -\hat{\mathbb{E}}_\mathcal{D}\left[\mathtt{CLIP}_{\hat{A}(x,y)}\left[\frac{\pi(y|x;\boldsymbol{\theta})}{\pi(y|x;\boldsymbol{\theta}_{\mathrm{old}})}\right]\hat{A}(x,y)\mathbb{1}_{\{Q_{1/3}(|\hat{A}|)<|\hat{A}(x,y)|<Q_{2/3}(|\hat{A}|)\}}\right], \tag{15}$$

$$\mathcal{L}_{\mathrm{ppo}}^\downarrow(\boldsymbol{\theta}) = -\hat{\mathbb{E}}_\mathcal{D}\left[\mathtt{CLIP}_{\hat{A}(x,y)}\left[\frac{\pi(y|x;\boldsymbol{\theta})}{\pi(y|x;\boldsymbol{\theta}_{\mathrm{old}})}\right]\hat{A}(x,y)\mathbb{1}_{\{|\hat{A}(x,y)|\leq Q_{1/3}(|\hat{A}|)\}}\right]. \tag{16}$$

**Remarks.** To avoid any misunderstanding, we briefly highlight the concepts of positive learning, negative learning, and loss reweighting. Regardless of specific formulations being minimized, we state that a part of an objective is doing positive learning if it increases the likelihood in generating data. Similarly, it performs negative learning if it decreases the corresponding likelihood (Zhang et al., 2024a; Zhu et al., 2025). Regarding loss reweighting, as a convention (Liu & Tao, 2015; Ren et al., 2018), we assume weight values to be non-negative, naturally satisfied by $\omega$ in DPO. However, in the context of PPO, the absolute of $\hat{A}$ determines the weight value. The sign is not of interest therein, but the focus of positive and negative learning instead. These rules lead to the separation from equation 6 to equation 10 in DPO and from equation 12 to equation 16 in PPO. Further please refer to Appendix E.2 for mathematical insights into the individual behaviors of these components.

**Learning Behaviors.** We report the *overall dynamics* in Figure 2(a), where $\mathcal{G}(\mathcal{L}_{\mathrm{ppo}};\boldsymbol{\theta}_{(t)})$ was evaluated per 400 steps. Overall, $\mathcal{G}$ keeps close to 0 but slightly negative, thus aligning with reinforcement learning behaviors. A closer look reveals two detailed aspects of this behavior. First, the exploration remains stable, as $\mathcal{G}$ keeps around 0 rather than oscillating between positive and negative extremes. Second, PPO explores relatively orthogonal yet mildly conflicting targets, with $\mathcal{G}$ tending to be slightly negative yet without extreme values. Some more intriguing aspects lie in identifying the key mechanisms that drive such exploration. We first examined the roles of *positive and negative learning*, monitoring $\mathcal{G}(\mathcal{L}_{\mathrm{ppo}}^+;\boldsymbol{\theta}_{(t)})$ and $\mathcal{G}(\mathcal{L}_{\mathrm{ppo}}^-;\boldsymbol{\theta}_{(t)})$ in Figure 2(b). Unlike DPO, we observed that positive learning is stable in shaping the learning targets, while negative learning keeps offsetting positive learning, thereby maintaining continued exploration.

When it comes to *loss reweighting*, as shown in Figure 2(c), we observed that the middle and top weighted components exhibit negative impacts on the final responses, with the top weighted data having a much stronger effect. On the other hand, $\mathcal{G}$ for the bottom weighted data is closer to 0, which is expected as their $|\hat{A}|$ values are close to 0, resulting in near-zero reweighting. To further comprehend the behaviors of the middle and top weighted components, we monitored the average raw advantages (instead of absolute advantages) for each weighted component in Figure 3. As we can see, the middle weighted data exhibit overall positive advantages. Along with their negative $\mathcal{G}$, this suggests that

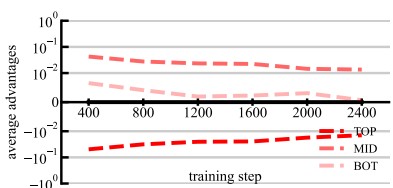

Figure 3: *Average (Raw) Advantages* during PPO for top (TOP), middle (MID), and bottom (BOT) weighted data.

data with positive advantages do not always dominate the learning directions. Instead, previously

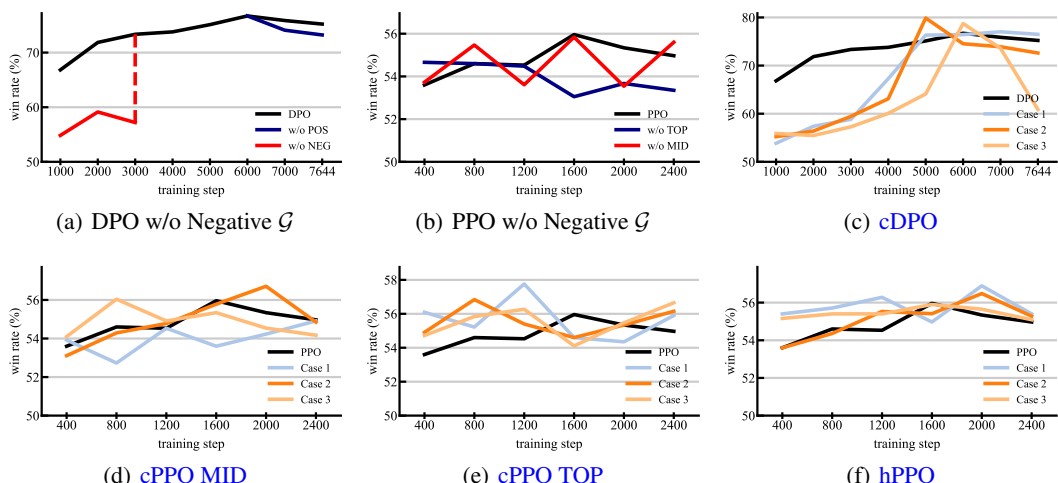

Figure 4: *Performance under Ablation*. For the Pythia-2.8B model trained on UltraChat 200K and tested on HH-RLHF-helpfulness, we show performance measured by Win Rate under: (a) DPO ablations removing negative learning (w/o NEG) before 3000 steps and positive learning (w/o POS) after 6000 steps, (b) PPO ablations removing top (w/o TOP) and middle (w/o MID) weighted data, (c) cDPO that emphasizes positive learning early and negative learning later, where we instantiate three representative dynamic-parameter settings: Case 1, Case 2, and Case 3 apply the cDPO coordination between training steps $(t_1, t_2) = (2500, 5500)$, $(2500, 6500)$, and $(3000, 8000)$ respectively, thereby balancing positive and negative learning, (d) cPPO that downweights top data where we examine varying degrees of downweighting: Case 1, Case 2, and Case 3 apply coefficients $\lambda = 0.3$, $\lambda = 0.7$, and $\lambda = 0.9$ to the top-weighted samples, respectively, , (e) cPPO that downweights middle data , where Case 1, Case 2, and Case 3 apply coefficients $\lambda = 0.7$, $\lambda = 0.5$, and $\lambda = 0.9$ to the middle-weighted samples, respectively, and (f) hPPO that changes learning behaviors periodically, where we vary the period $t_3$ and amplitude $\tau$ across three representative settings: Case 1, Case 2, and Case 3 correspond to $(t_3, \tau) = (2, 0.08)$, $(5, 0.05)$, and $(20, 0.01)$, respectively. We present illustrative results here, with additional results and best performance in Appendix D.

achieved positive reward actions may be superseded by newly explored ones. For the top weighted data, it appears that their negative $\mathcal{G}$ straightforwardly stems from the overall negative advantages, indicating that these tokens are overall dispreferred and thus unlearned.

**Summary.** Now, it is the time to answer "What is PPO doing, how and why?" From our aspect of gradient dynamics, PPO is reaffirmed to be reinforcement learning. Moreover, the exploration encourages coverage over a broad range of slightly conflicting responses. As an online method, positive and negative learning maintain stable roles. Positive learning guides the model in identifying new, competitive targets, while negative learning counterbalances its impacts to foster further exploration. Moreover, loss reweighting, as reflected by the absolute advantages, demonstrates that different ranges of weighted data serve distinct roles. The top weighted component is more likely to contain dispreferred data with negative advantages that should be unlearned, as in $\mathcal{L}_{\text{ppo}}^-$. In contrast, middle weighted components are more likely to contain locally preferred data with positive advantages, which will not be reinforced but instead suppressed by new explorations. This interaction may be one of the factors that facilitate the effective exploration of PPO.

## 4    FURTHER ANALYSES: FROM THE PERFORMANCE ASPECT

One might simply think that removing certain components from the objective with negative $\mathcal{G}$, which harm the final responses, would accelerate convergence and improve performance. Unfortunately, this is not strictly correct, as removing certain components alters the overall learning dynamics and will yield new final responses. The improved performance for this new $\mathcal{D}'$ is not guaranteed, and the dynamics of other preserved operations are also affected. Therefore, from the aspect to improve performance, our analyses above primarily serve as motivating insights, while empirical validations

are further required for their practical effectiveness. Hence, this section poses a series of questions, each answered and supported by experiments with performance measurements. We keep this section concise for clarity, deferring implementations, hyper-parameters, and other results to Appendix D.

**Do components with negative $\mathcal{G}$ hurt performance?** A negative $\mathcal{G}$ by definition harms the final responses. Yet, it is unclear whether such components will degrade performance or act as regularizers. To study this, we take Figure 1(b) and Figure 2(c) as examples, removing components with negative $\mathcal{G}$ and comparing the performance to their original versions, measured by Win Rate. In Figure 4(a), we removed negative learning for DPO during the first 3000 steps, and another variant continued from the 6000-step DPO checkpoint but excluded positive learning. In Figure 4(b), we removed either the top or the middle weighted data for PPO throughout training. As observed, removing components with negative $\mathcal{G}$ leads to decreased performance and/or unstable optimization, indicating that *these components are beneficial and act more like regularizers*. Moreover, Figure 4(b) further supports the distinct roles of data within different weighting ranges in PPO, as discussed in Section 3.2. Specifically, middle weighted data that are locally high-rewarding contribute to stable optimization, and their removal induces instability. By contrast, top weighted data are overall dispreferred, and removing them hinders the model from further exploration, thus resulting in degraded performance.

**Could fine-grained control yield improvements?** We have identified components within DPO and PPO that exhibit regularization effects, raising the question of whether their careful control would lead to further improvements. In Appendix D, we introduce two variants: *controlled DPO* (cDPO), initially emphasizing positive learning and gradually shifting to negative learning in DPO, and *controlled PPO* (cPPO), which downweights either the middle or top weighted data in PPO. The results are presented in Figures 4(c)-4(e). We observed many cases with the improved performance after applying these controls, evidenced by comparisons of either the peak performance during training or the final performance. This suggests that *PO methods could benefit from fine-grained, component-level controls*, particularly from the perspective of regularization. Note that in Figure 4(c), negative learning largely dominates the learning dynamics during the later training phase, followed by notable performance degradation in Cases 2 and 3. This indicates that, while negative learning is more beneficial at this stage, positive learning remains essential to prevent model collapse. Moreover, comparing Figures 4(d) and 4(e), we observed that downweighting top weighted data can lead to better performance than that for middle weighted data. This aligns with cases where middle-weighted data, which are locally high-rewarding, are preserved, while the strength of exploration associated with top-weighted data is slightly reduced. However, due to a trade-off between exploration and exploitation, its degree should be carefully controlled, cf., Appendix D.

**Could PO methods benefit from changing behaviors?** In PPO, positive learning drives the model toward high-reward targets, while negative learning counteracts this tendency to promote exploration. By slightly weakening negative learning, we can shift the overall behavior toward supervised learning. In Appendix D, we propose *hybrid PPO* (hPPO), where the strength of negative learning is periodically reduced, causing the learning process to alternate between reinforcement and supervised learning. Figure 4(f) presents a comparison with the original PPO. We found that *hPPO can achieve notable improvements over the original PPO*. However, as detailed in Appendix D, this improvement occurs only when the duration of supervised learning is short and the degree of downweighting is tiny. Otherwise, the model would risks overfitting to locally high-reward actions.

## 5 RELATION TO PREVIOUS WORKS

The continued advancement of the field is built on the foundations of DPO and PPO, enriched by numerous new techniques and standards. We situate our findings within this broader context to offer a fresh viewpoint on recent achievements, further substantiating our contributions and generality.

Many code repositories suggested *gradient clipping* by default, limiting the gradient norm to prevent exploding gradients. In Appendix C, we observed that such extreme cases are rare; however, when they do occur, they typically result in a large negative overall $\mathcal{G}$. In contrast, removing them yields an overall $\mathcal{G}$ that is near zero in PPO and positive in DPO. This observation suggests that such outliers can exert disproportionately large negative effects on the stability of optimization dynamics, potentially disrupting information accumulated over many prior training steps.

Moreover, beyond the dynamic learning rate inherently provided by Adam (Kingma & Ba, 2014) and AdamW (Loshchilov & Hutter, 2017), DPO typically employs an additional *decayed learning rate scheduler*, whereas PPO generally does not. This practice implies an implicit opinion within the community: DPO is supervised learning that can converge within a limited number of training steps, whereas PPO is reinforcement learning, where the point of achieving a proper policy is uncertain.

*Simple preference optimization* (SimPO) (Meng et al., 2024) proposes replacing the reward formulation in DPO with the average log-likelihood, resulting in a simpler objective that eliminates the reference model. Following the derivation similar to equation 5, SimPO also involves *loss reweighting*, but takes the form resembling $\sigma[\log \pi(y^-|x; \boldsymbol{\theta}) - \log \pi(y^+|x; \boldsymbol{\theta})]$. This indicates that SimPO employs a reweighting function that contrasts with the implicit reward mechanism $\omega(x, y^+, y^-)$ in DPO, yet yields the same effect of assigning larger weights to data with insufficient margins.

A previous paper (Saeidi et al., 2024) shows that *Kahneman-Tversky optimization* (KTO) (Ethayarajh et al., 2024) and *contrastive Preference Optimization* (CPO) (Xu et al., 2024), which are variants of DPO, can bypass the pre-processing step of SFT, which is typically indispensable for the original DPO. From the aspect of positive and negative learning, we observe that they place *greater emphasis on positive learning*—via larger weights in KTO and an additional cross-entropy term on win data in CPO. This suggests that the learning dynamics are already dominated by positive learning at the beginning, thereby mitigating the need of model pre-processing.

*Group relative policy optimization* (GRPO) (Shao et al., 2024) introduces *reward normalization*, which normalizes rewards within a group of responses for each question. This ensures that an equal number of responses contribute to both positive and negative learning. Therefore, the gradients for positive and negative learning are approximately balanced, leading to an overall near-zero $\mathcal{G}$. Consequently, exploration behaviors can be stably maintained throughout its learning process.

*Decoupled clip and dynamic sampling policy optimization* (DAPO) (Yu et al., 2025), along with other practitioners, advocates a *clip-higher* strategy in PPO-based methods, which assigns a larger clipping threshold $\epsilon$ for positive $\hat{A}$. From the gradient perspective, such asymmetry allows positive learning to contribute stronger gradient magnitudes than negative ones, thereby steering slightly strong learning tendency towards positive advantages. Potentially, using clip-higher can converge faster by leveraging short-term high-reward targets explored early during positive learning, albeit reducing exploration.

Some of these above methods are explored in the context of reasoning rather than preference alignment. These connections indicate that our findings are not confined to preference alignment alone but may further contribute to the broader areas that also rely on PO, which we aim to explore in the future.

## 6 CONCLUSIONS

We examined the learning behaviors of PO methods through their optimization dynamics in achieving the final responses, focusing on DPO and PPO under preference optimization as case studies. We show in Section 3 that these methods exhibit different optimization dynamics, with distinct components contributing in different ways to shape their overall learning behaviors. Building on these findings, we motivated strategies to further control the behaviors of the original DPO and PPO in Section 4, offering deep insights and the potential for improved performance. These discussions help explain the success of many latest methods and highlight opportunities for future advances in Section 5. We further envision the opportunity of *coordinated preference optimization (CPO)*. Our central position is that the next generation of PO methods should carefully monitor and coordinate the influence of individual components. CPO would involve two parallel pipelines: one for training the main model with specific combinations of basic components, and another for sampling checkpoints of the training model to analyze the real-time impact of each component. Using this feedback, we can adjust the roles of each component to obtain a more desirable combination that adapts to the current stage of model training, which we will explore in future work.

## ETHICS STATEMENT

All authors have read and adhered to the ICLR Code of Ethics. Our study relies solely on publicly available datasets and models, as detailed in Appendix B. No private or personally identifiable

information was used. The work aims to advance the scientific understanding of PO methods while upholding principles of transparency, fairness, and responsible research.

## REPRODUCIBILITY STATEMENT

We provide an anonymous repository at `https://anonymous.4open.science/r/WHAT-IS-PREFERENCE-OPTIMIZATION-DOING-HOW-AND-WHY-96EFJ3E`, which contains our source code, experimental configurations, and evaluation scripts. The codebase will be made publicly available upon acceptance. All base models and PO benchmarks used in this work are publicly accessible, as detailed in Appendix B. All experiments were conducted using NVIDIA A100-80GB GPUs with Python 3.11 and PyTorch 2.4.1.

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

## A  FURTHER DISCUSSIONS

Let us first summarize our gradient-based analysis framework. We aimed to identify whether the optimization dynamic of a particular PO method align with supervised or reinforcement learning, signified by the stability during its learning in achieving the final responses. This goal can be achieved by monitoring the gradient dot products between the learning objective and the expected negative log-likelihood over the final responses, formulated as in equation 3. The exact PO behavior, either following supervised or reinforcement learning, can then be identified based on the following.

- *Supervised learning* produces consistently positive values of $\mathcal{G}$ during training, indicating stable learning targets that align with the directions of the final responses.
- *reinforcement learning* yields near-zero, negative, or fluctuating $\mathcal{G}$, reflecting exploratory behaviors that pursue changing targets that may fail to benefit the final responses.

**Contributions.** While PO behaviors have more or less been discussed, few studies have attempted to verify them from the perspective of optimization dynamics. We fill this gap by analyzing gradient behaviors, employing a first-order approximation to make the overall framework more actionable. The gradient-based analysis is a well-established and widely studied approach (Pruthi et al., 2020; Koh & Liang, 2017). However, our focus differs substantially from these previous works. First, we focus on generative models, whereas previous works mainly studied discriminative models. This shift introduces new challenges, as PO objectives may involve implicit targets and lack explicit ground truths. Second, previous works typically emphasize the data perspective in their analysis, but we further explore the objective (cf., Section 3) and the model perspectives (cf., Appendix C). Our analysis uncovers some intriguing findings that are rarely discussed in existing literature, offering deeper insights and potentially inspiring future research.

**Limitations.** We acknowledge our limitations. First, Adam (Kingma & Ba, 2014) or its variants are commonly used for LLM training instead of vanilla SGD. Its adaptive step size and averaged momentum introduce non-linearities, which are not accounted for in our first-order analysis. Moreover, the non-convexity of the model landscape makes it hard to guarantee of monotonic loss decreasing (Zhang et al., 2024b). During some training steps, loss values may increase, where a probed negative $\mathcal{G}$ may better indicate that the PO objective is contributive at theses points. For simplicity, we defer addressing the first limitation of high-order analysis to future work and manually filter out training steps with decreased $\mathcal{L}$ when computing $\mathcal{G}$ to address the second limitation. Moreover, the sampling intervals are relatively coarse and may fail to capture finer-grained variations in training sensitivity. Future work could benefit from extending training durations and adopting smaller checkpoint intervals.

A remark, rather than a limitation, is that the gradient alignment condition in equation 3 focuses on measuring the stability of the targets, rather than evaluating performance changes. Likelihood estimation and fitted reward functions in practice are biased indicators of true performance (Kwon et al., 2023), which should instead be assessed by human experts or through LLM-as-a-judgment. Sadly, these gold evaluation pipelines are not differentiable, thus hard to integrate into our gradient analysis. On the other side, since hyper-parameters have been tuned for each alignment method, we ensure that increasing the likelihood for data within $\mathcal{D}'$ leads to improved performance over that before alignment. From such an indirect and limited way, we ensure that our analyses can also somehow signify performance changes. The real problem behind this remark is that when we find improvements that appear to modify the original learning behaviors for the better, we must rerun these improved methods from scratch and reevaluate their performance. This is because the targets have changed, and we cannot guarantee that these new targets are superior definitely.

## B  EXPERIMENTAL CONFIGURATIONS

In this section, we provide additional descriptions of our implementation details, covering training setups, gradient alignment estimations, as well as performance evaluation protocols.

### B.1  PO TRAINING SETUPS

Following (Tunstall et al., 2023), we first pre-processed all base models with SFT on the UltraChat-200k dataset, and then performed PO training using the UltraFeedback dataset. Our implementations

were built on the *transformers reinforcement learning* library[1], and the detailed configurations for DPO and PPO are presented as follows.

**DPO.** For Pythia-2.8B, we set the temperature parameter to $\beta = 0.1$, a widely adopted choice (Rafailov et al., 2024), which helps control the deviation from the reference model for stable training. The base model was trained for 2 epochs using the AdamW optimizer with a global batch size of 32, an initial learning rate of $5 \times 10^{-7}$, and a weight decay of 0.01. Moreover, we employed a linear learning rate schedule with a 5% warmup ratio and applied gradient clipping with a maximal-allowed gradient norm to be 1. For Qwen3-1.7B, we set the temperature parameter to $\beta = 0.1$ and trained the model for two epochs using the AdamW optimizer with a global batch size of 512, an initial learning rate of $5 \times 10^{-5}$, and a weight decay of 0.01. For Llama3-8B, we use the same objective with $\beta = 0.1$ and train the model for two epochs using AdamW with a global batch size of 256, an initial learning rate of $1 \times 10^{-5}$, and a weight decay of 0.01.

**PPO.** For Pythia-2.8B, the base model was trained with a global batch size of 128 and an initial learning rate of $2 \times 10^{-6}$ with the AdamW optimizer. No learning rate schedule was applied, in accordance with common practice, but gradient clipping was employed with a maximum norm of 1. For candidate roll-outs, we sampled a single response using top-$p = 0.9$ and temperature 1.0, truncating up to 400 tokens. A scalar reward was then assigned by ArmoRM-Llama3-8B-v0.1 (Wang et al., 2024), an off-the-shelf reward model trained on human preference data. For Qwen3-1.7B, we trained the model with a global batch size of 128 and an initial learning rate of $2 \times 10^{-6}$ with the AdamW optimizer. No learning rate schedule was applied, and gradient clipping was enforced with a maximum norm of 1. For rollouts, we generated one response per prompt using top-$p = 0.9$ and temperature 0.7, and a maximum generation length of 400 tokens, while constraining the prompt length to 256 tokens. For Llama3-8B, we trained the model with a global batch size of 32 and an initial learning rate of $5 \times 10^{-7}$ with the AdamW optimizer. No learning rate schedule was applied, and gradient clipping was enforced with a maximum norm of 1. For rollouts, we generated one response per prompt using top-$p = 0.9$ and temperature 0.7, and a maximum generation length of 400 tokens, while constraining the prompt length to 256 tokens. For both Pythia-2.8B and Qwen3-1.7B, a scalar reward was assigned using ArmoRM-Llama3-8B-v0.1 (Wang et al., 2024), while Llama3-8B used Skywork-Reward-Llama3.1-8B (Liu et al., 2024b) for reward computation.

For proposed controlled variants, such as cDPO, cPPO, and hPPO, we adopted the same datasets, base models, and hyperparameter configurations as their baseline counterparts. This ensures a strictly controlled comparison, so that any observed differences in performance can be attributed solely to the algorithmic modifications introduced by the proposed objectives.

### B.2 GRADIENT ALIGNMENT SETUPS

To reduce the costs of gradient computations, we sampled 500 data points from $\mathcal{D}$ and $\mathcal{D}'$ respectively when computing $\mathcal{G}$. To ease GPU memory usage, we set the batch size to 4 in DPO and 6 in PPO when partitioning both $\mathcal{D}$ and $\mathcal{D}'$. Moreover, to align with the common practice of gradient clipping, we excluded data batches with extreme gradient magnitudes using the *inter-quartile range* (IQR), a robust strategy of outlier detection that measures the variability within the value distribution. Concretely, given the distribution of gradient norms across mini-batches, we compute the first quartile $(Q_1)$ and the third quartile $(Q_3)$, and define an outlier threshold as $Q_3 + 1.5 \times (Q_3 - Q_1)$. Any mini-batch whose gradient norm exceeds this threshold is regarded as an outlier and excluded from the computation of $\mathcal{G}$.

### B.3 PERFORMANCE EVALUATION SETUPS

We assessed PO performance using an LLM-as-a-judge framework on AlpacaEval (Dubois et al., 2024), with responses generated under the greedy decoding and constrained to a maximum length of 512 tokens. The evaluation metric, *Win Rate*, as convention, is defined as the proportion of prompts for which the judge model prefers the output of a candidate model over that of the SFT baseline.

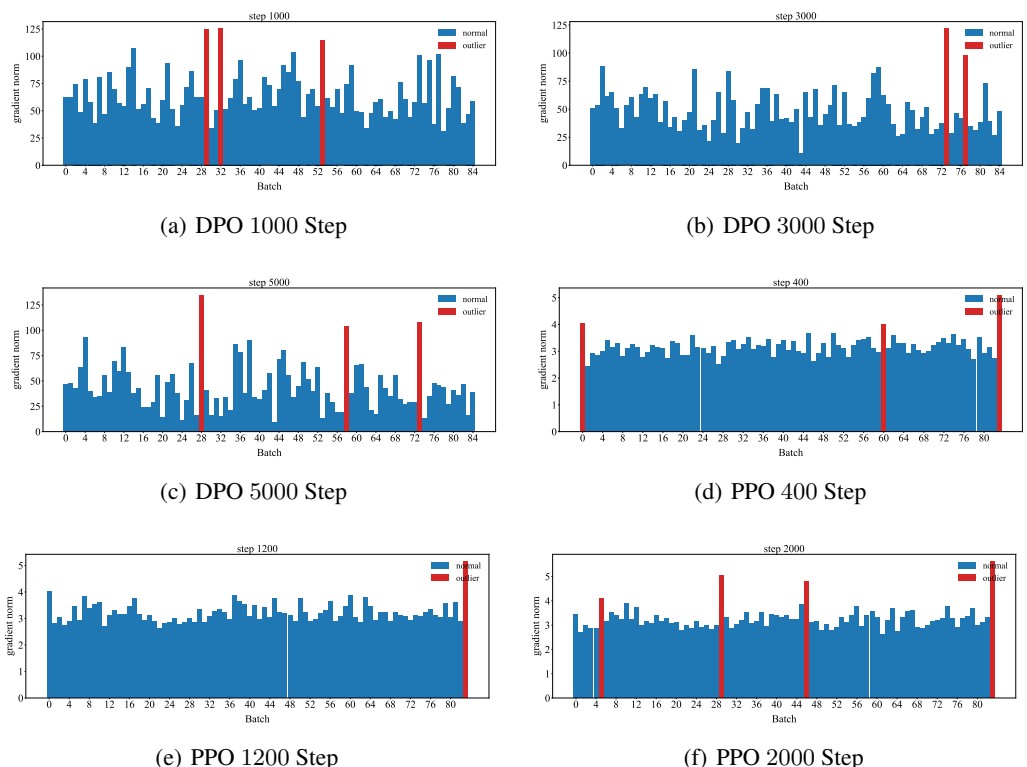

Figure 5: *Gradient Magnitudes*. For the Pythia-2.8B model trained on UltraChat 200K and tested on HH-RLHF-helpfulness, we illustrate the distributions of gradient magnitudes computed with respect to mini-batches for DPO and PPO, across training steps. Normal data points are colored in blue, while outliers detected by IQR are colored in red.

## C SUPPORTING EXPERIMENTS

We conducted further experiments to address additional questions of interest, which will substantiate the assumptions presented in the main text and offer new insights.

**Why is gradient clipping typically adopted in PO methods?** Many codebases recommend gradient clipping when using PO methods, which constrains the gradient norm when it exceeds a predefined threshold. To emulate this practice, we have discarded those batches whose gradient norms are overly large when computing $\mathcal{G}$. This seemingly minor detail leads to notable changes in observed learning behavior. First, in Figure 5, we illustrate for both DPO and PPO, outlier cases with extreme gradient norms (colored in red) indeed occur. However, these cases constitute only a minority relative to the normal data (colored in blue) throughout training. Then, in Figure 6, we compare gradient alignment with and without outlier filtering. The influence of gradient outliers is stronger for DPO than for PPO, notably changing the behaviors of the former. It may be explained by the higher frequency of outlier cases in DPO and suggests that gradient clipping is more crucial for stabilizing DPO than PPO.

**Why do the roles of positive and negative learning exchange in DPO?** In Section 3.1, we demonstrated that positive learning contributes to shaping the targets at the beginning but becomes harmful in the later training phase, from the perspective in acheiving the final responses. In contrast, negative learning is detrimental during the early phase but becomes beneficial as training progresses. A seemingly simple explanation for this phenomenon is that the overall gradient initially tends to positive learning, encouraging $-\hat{\mathbb{E}}_{\mathcal{D}}\,\omega(x, y^+, y^-) \log \pi(y^+|x; \boldsymbol{\theta}^{(t)})$ toward converge with diminishing gradient magnitudes. At the same time, the gradient associated with negative learning, moving in a relatively opposite direction, tends to diverge with increasing magnitudes. As training progresses, the

---

[1] https://huggingface.co/docs/trl

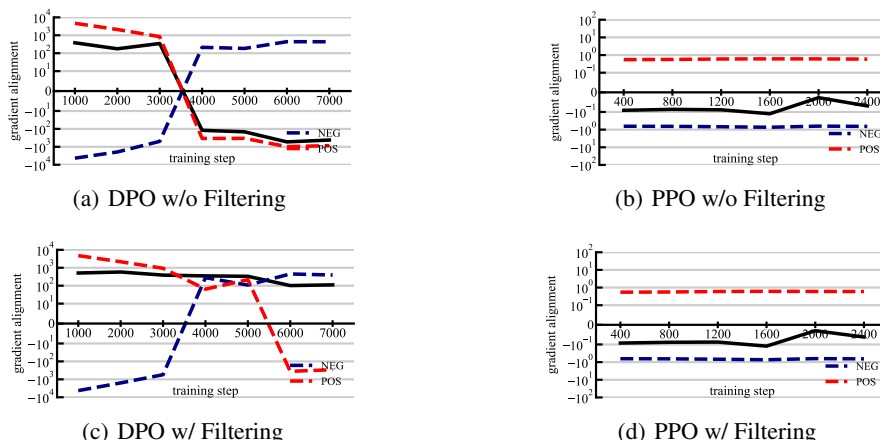

Figure 6: *Gradient Dynamics With and Without Outlier Filtering*. For the Pythia-2.8B model trained on UltraChat 200K and tested on HH-RLHF-helpfulness, we present the learning dynamics of $\mathcal{G}$ for (a) DPO and (b) PPO without excluding batches with extreme gradient magnitudes, in contrast to the main results with outlier filtering shown in (c) and (d), as reported in the main text.

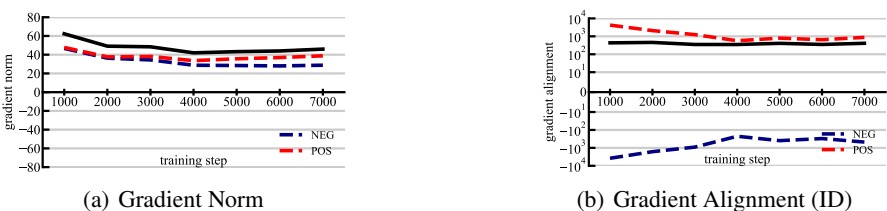

Figure 7: *DPO Gradient Dynamics*. For the Pythia-2.8B model trained on UltraChat 200K and tested on HH-RLHF-helpfulness, we report (a) gradient magnitudes and (b) gradient alignments for DPO. Here, $\mathcal{D}'$ is built from the training dataset, so we focus on in-distribution rather than out-of-distribution responses as in Figure 1(b) of the main text.

learning targets become dominated by negative learning. However, in Figure 7(a), we observed that this assumption does not hold, as the gradient magnitude of positive learning is consistently larger than that of negative learning throughout DPO.

It motivates our assumption that overfitting to $y^+$ is the primary reason for their exchanged roles. To justify this, we constructed another final response dataset from training data when computing the gradient alignment condition. This means that when constructing $\mathcal{D}'$, we used the UltraChat 200K dataset (the training set) rather than HH-RLHF-helpfulness. Therefore, our analysis focuses on in-distribution responses rather than out-of-distribution generalization. We show the corresponding results in Figure 7(b). Unlike the out-of-distribution responses in Figure 1(b), the the scenario of exchanging roles disappears, supporting our assumption that overfitting is a reasonable explanation.

**Why is SFT typically used as model pre-processing?** Parameter initialization is crucial, where base models are typically pre-processed by SFT before PO training. For DPO, Figures 8(a)–8(b) show that positive and negative gradients are of the same order with or without SFT, and that positive gradients consistently dominate negative ones, offering limited insights in explaining the importance of SFT. Instead, Figure 8(c) shows that without SFT, final responses are dominated by positive learning, with limited influence from negative learning. We therefore conjecture that SFT enables DPO to overfit $y^+$ easily, then allowing negative learning to guide target learning. For PPO, as mentioned in Section 3, while it can facilitate the exploration of slightly conflicting answers, this strength is limited, and the model may require long time to find proper responses. Therefore, pre-processed models with SFT help confine the search space for PPO, serving as a strong start point for its training success.

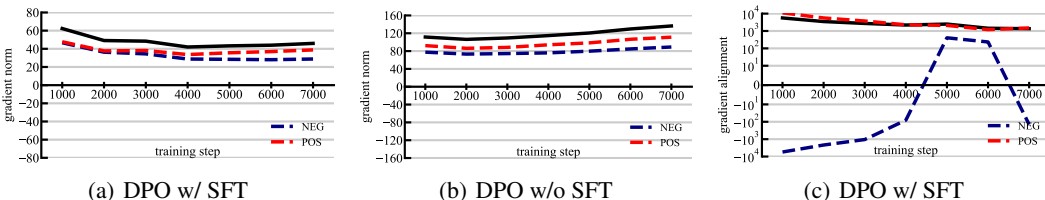

Figure 8: *DPO Gradient Dynamics*. For the Pythia-2.8B model trained on UltraChat 200K and tested on HH-RLHF-helpfulness, we compare DPO with and without SFT, reporting gradient magnitudes (a) with SFT, (b) without SFT, and (c) gradient dynamics without SFT.

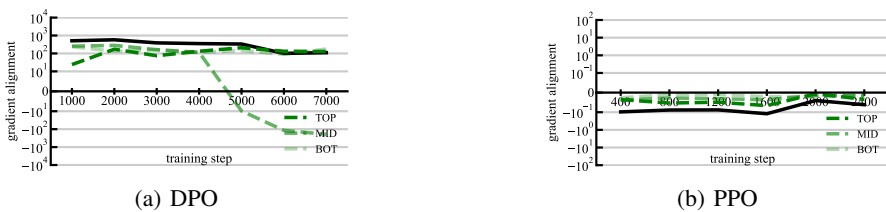

Figure 9: *Layer-wise Learning Dynamics*. For the Pythia-2.8B model trained on UltraChat 200K and tested on HH-RLHF-helpfulness, we report the layer-wise dynamics for (a) DPO and (b) PPO.

**Do PO methods affect different layers with distinct behaviors?** Our analysis framework, based on gradient dot products, is very flexible to support beyond component-level analysis. For example, we can examine the impacts of PO methods for particular model layers of interest, which is a layer-level analysis. In Figure 9, we partitioned the model into three segments: bottom layers (closer to the inputs), middle layers, and top layers (closer to the outputs), assessing their gradient alignment conditions respectively. As shown, the layer-wise curves resemble the overall behavior of $\mathcal{G}$, expect for the middle layers in DPO. The resemblance suggests that training only a part of the model, e.g., the top layers, may suffice to achieve comparable performance. The exceptional behavior observed in the middle layers for DPO can be attributed to the stronger influence of negative learning on these layers, a phenomenon also reported in (Wang et al., 2025).

**Go Beyond Pythia.** We used Pythia-2.8B as the base model because it is open-sourced with detailed documentation of its training and fine-tuning procedures, making it well-suited for rigorous verification. For instance, we know that this base model has not been trained with any PO procedures or exposed to any PO datasets, thereby eliminating potential confounding factors. However, it is also important to explore a broader range of models to further strengthen the generality of our conclusions. Therefore, we additionally tested the gradient alignment conditions on Qwen3-1.7B, with the results summarized in Figure 10. Similar observations can be drawn from Qwen3-1.7B as from Pythia-2.8B. Overall, DPO still exhibits supervised learning behavior, while PPO exhibits reinforcement learning behavior. For DPO, both positive and negative learning contribute to shaping the learning targets, whereas for PPO, positive learning shapes the targets while negative learning instead supports exploration. Moreover, top- and middle-weighted data in PPO contribute more than low-weighted data, while this does not hold for DPO. These findings align with our results in Figure 1 and Figure 2 for Pythia-2.8B, suggesting that our conclusions have the potential to generalize across a broad range of model families. We further provide ablation studies for Qwen3-1.7B in Figure 11, echoing Figure 4 for Pythia-2.8B. As observed, Qwen3-1.7B exhibits similar behaviors for DPO and PPO without negative $\mathcal{G}$ as those seen for Pythia-2.8B. Furthermore, more fine-grained control of DPO and PPO, following cDPO, cPPO, and hPPO, leads to further improvements over the baselines, and these improvements are even more pronounced than those for Pythia-2.8B.

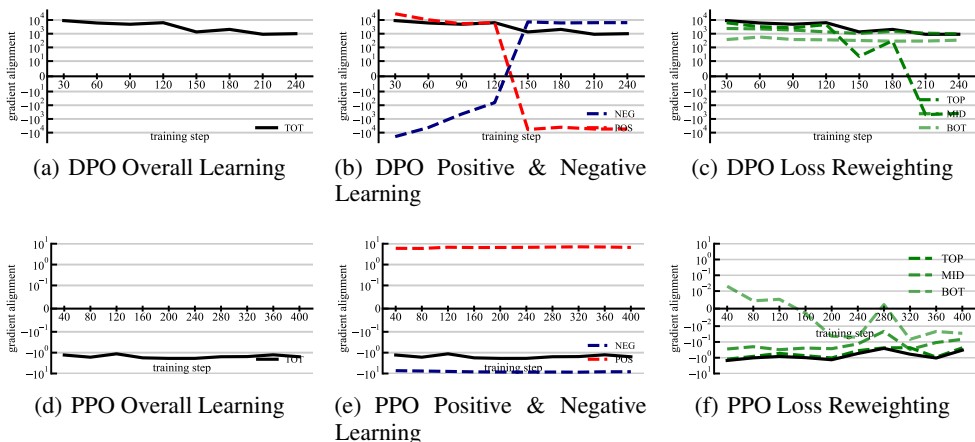

(a) DPO Overall Learning

(b) DPO Positive & Negative Learning

(c) DPO Loss Reweighting

(d) PPO Overall Learning

(e) PPO Positive & Negative Learning

(f) PPO Loss Reweighting

Figure 10: *Qwen Learning Dynamics*. For the Qwen3-1.7B model trained on UltraChat 200K and tested on HH-RLHF-helpfulness, we show the learning dynamics of $\mathcal{G}$ for DPO and PPO, covering the overall objectives ((a) for DPO and (d) for PPO), the positive and negative components ((b) for DPO and (e) for PPO), and each weighted component ((c) for DPO and (f) for PPO), respectively.

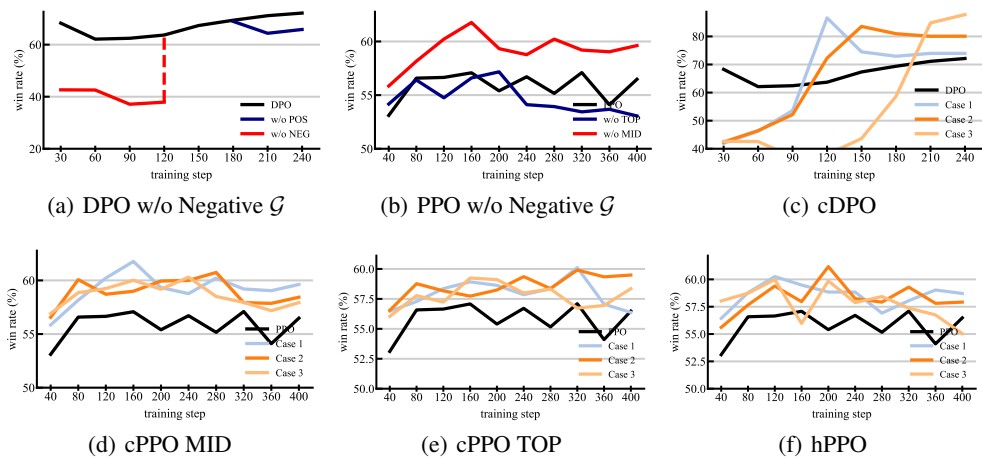

(a) DPO w/o Negative $\mathcal{G}$

(b) PPO w/o Negative $\mathcal{G}$

(c) cDPO

(d) cPPO MID

(e) cPPO TOP

(f) hPPO

Figure 11: *Performance under Ablation*. For the Qwen3-1.7B model trained on UltraChat 200K and tested on HH-RLHF-helpfulness, we show performance measured by Win Rate under: (a) DPO ablations removing negative learning (w/o NEG) and positive learning (w/o POS), (b) PPO ablations removing top (w/o TOP) and middle (w/o MID) weighted data, (c) cDPO that emphasizes positive learning early and negative learning later, where we instantiate three representative dynamic-parameter settings: Case 1, Case 2, and Case 3 apply the cDPO coordination between training steps $(t_1, t_2) = (40, 160)$, $(0, 240)$, and $(120, 240)$ respectively, thereby balancing positive and negative learning, (d) cPPO that downweights top data where we examine varying degrees of downweighting: Case 1, Case 2, and Case 3 apply coefficients $\lambda = 0.0$, $\lambda = 0.3$, and $\lambda = 0.5$ to the top-weighted samples, respectively, (e) cPPO that downweights middle data, where Case 1, Case 2, and Case 3 apply coefficients $\lambda = 0.7$, $\lambda = 0.9$, and $\lambda = 0.5$ to the middle-weighted samples, respectively, and (f) hPPO that changes learning behaviors periodically, where we vary the period $t_3$ and amplitude $\tau$ across three representative settings: Case 1, Case 2, and Case 3 correspond to $(t_3, \tau) = (2, 0.01)$, $(20, 0.01)$, and $(50, 0.01)$, respectively.

## D    Component Coordination

In Section 4, we evaluated several variants of DPO and PPO, revealing their potential improvements from a regularization perspective. Here, we present additional implementation details and results with other hyper-parameter configurations, aiming to offer deeper insights.

**Controlled DPO and PPO**. We have demonstrated the regularization effects for those components within PO learning objectives, motivating Section 4 to investigate whether their fine-grained controls can yield further improvement. For DPO, we have shown that negative learning in the early phase and positive learning in the later phase act as regularizers. Therefore, we introduce a dynamic parameter $\lambda = \max\left(\min\left(\frac{t-t_1}{t_2-t_1}, 1\right), 0\right)$, which monotonically increases from 0 to 1 between $t_1$ and $t_2$ training steps, to adjust the balance between positive learning and negative learning. Accordingly, we generalize DPO and propose *controlled DPO* (cDPO), which is given by

$$\mathcal{L}_{\text{cdpo}}(\boldsymbol{\theta}) = -\hat{\mathbb{E}}_{\mathcal{D}}\left[\log\sigma\left(\beta\left[(1-\lambda)\log\frac{\pi(y^+|x;\boldsymbol{\theta})}{\pi(y^+|x;\boldsymbol{\theta}^{\text{ref}})} - \lambda\log\frac{\pi(y^-|x;\boldsymbol{\theta})}{\pi(y^-|x;\boldsymbol{\theta}^{\text{ref}})}\right]\right)\right] \quad (17)$$

As $\lambda$ increases, cDPO starts with positive learning and gradually shifts toward negative learning. This allows the strength of the regularization effect to be modulated in a flexible manner.

We present results in Figures 12-13. As observed, applying positive learning alone yields only limited performance improvement, i.e., steps before $t_1$. Then, the gradual incorporation of positive learning, i.e., steps after $t_1$, leads to further gains, highlighting the important role of negative learning in enhancing performance. However, when the influence of negative learning overwhelms that of positive learning, i.e., steps close to or after $t_2$, performance typically degrades. It indicates that negative learning alone is detrimental, and its effectiveness relies on the regularization effect of positive learning. However, we truly observed many cases where the best performance surpasses that of the original DPO, suggesting that coordinating the strengths of positive and negative learning offers a promising way to improve DPO, thus warranting further studies.

Moreover, for PPO, we also introduce its controlled variant, termed *controlled PPO* (cPPO). As discussed in Section 4, the idea is to selectively downweight data depending on the range of their advantages. Specifically, we consider two variants: downweighting the top weighted samples, i.e.,

$$\mathcal{L}_{\text{cppo}}^{\text{top}}(\boldsymbol{\theta}) = -\hat{\mathbb{E}}_{\mathcal{D}}\left[\text{CLIP}_{\hat{A}(x,y)}\left[\frac{\pi(y|x;\boldsymbol{\theta})}{\pi(y|x;\boldsymbol{\theta}_{\text{old}})}\right]\hat{A}(x,y)\right.$$
$$\left.\left[\lambda\mathbb{1}_{\{|\hat{A}(x,y)|\geq Q_{2/3}(|\hat{A}|)\}} + \mathbb{1}_{\{|\hat{A}(x,y)|<Q_{2/3}(|\hat{A}|)\}}\right]\right], \quad (18)$$

and downweighting the middle weighted samples, i.e.,

$$\mathcal{L}_{\text{cppo}}^{\text{mid}}(\boldsymbol{\theta}) = -\hat{\mathbb{E}}_{\mathcal{D}}\left[\text{CLIP}_{\hat{A}(x,y)}\left[\frac{\pi(y|x;\boldsymbol{\theta})}{\pi(y|x;\boldsymbol{\theta}_{\text{old}})}\right]\hat{A}(x,y)\right.$$
$$\left.\left[\mathbb{1}_{\{|\hat{A}(x,y)|\geq Q_{2/3}(|\hat{A}|)\}} + \mathbb{1}_{\{|\hat{A}(x,y)|\leq Q_{1/3}(|\hat{A}|)\}} + \lambda\mathbb{1}_{\{Q_{2/3}(|\hat{A}|)>|\hat{A}(x,y)|>Q_{1/3}(|\hat{A}|)\}}\right]\right], \quad (19)$$

where $0 \leq \lambda \leq 1$ is a predefined hyper-parameter controlling the degree of downweighting.

The results in comparing between cPPO and PPO are summarized in Figures 14-16. When controlling either middle or top weighted data, learning tends to be slightly less stable but shows potential for improvement. Between the two setups, controlling top-weighted data yields better results and exhibits lower sensitivity to hyper-parameters. These findings suggest that downweighting top weighted samples in cPPO is a more promising approach for enhancing PO.

**Hybrid PPO.** This fact facilitates us to explore the possibility to make PPO alternate between these two stages, leading to *hybrid PPO* (hPPO) as

$$\mathcal{L}_{\text{hppo}}(\boldsymbol{\theta}) = -\hat{\mathbb{E}}_{\mathcal{D}}\left[\text{CLIP}_{\hat{A}(x,y)}\left[\frac{\pi(y|x;\boldsymbol{\theta})}{\pi(y|x;\boldsymbol{\theta}_{\text{old}})}\right]\hat{A}(x,y)\left[\mathbb{1}_{\{\hat{A}(x,y)\geq 0\}} + \lambda\mathbb{1}_{\{\hat{A}(x,y)<0\}}\right]\right], \quad (20)$$

where $\lambda = \max\left(\min\left(\sin\frac{\pi t}{t_3}, 0\right), -\tau\right) + 1$ and $\tau > 0$. This scheduler of $\lambda$ makes training alternate between reinforcement and supervised learning: it begins with the reinforcement stage, switches to

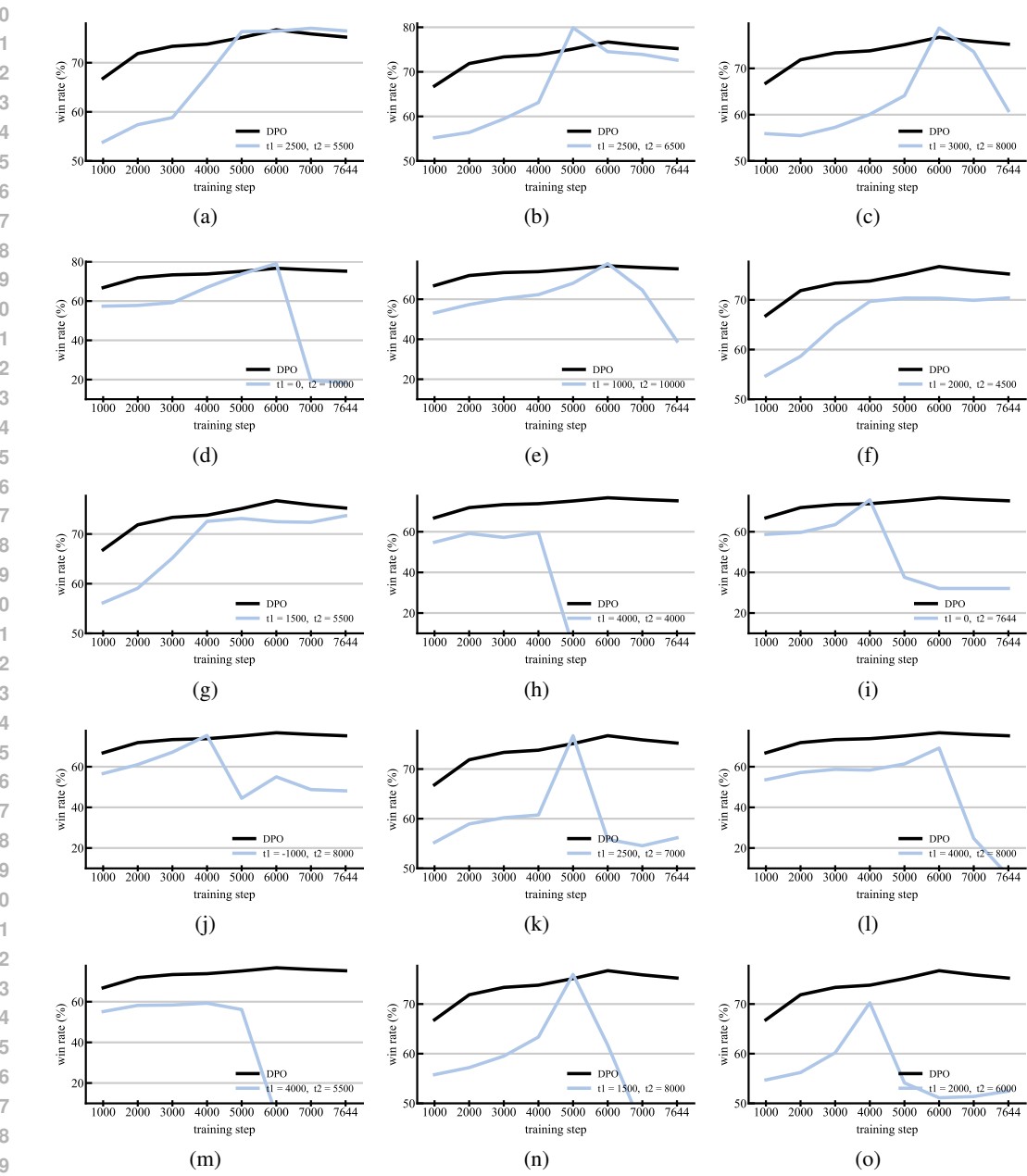

Figure 12: *DPO vs. cDPO*. For the Pythia-2.8B model trained on UltraChat 200K and tested on HH-RLHF-helpfulness, we compare DPO with cDPO across hyper-parameters.

supervised after $t_3$ steps, and returns to reinforcement gradually after another $t_3$ steps, repeating with the period $2 \times t_3$. We present the results in Figures 17-18. We first observed that hPPO performs well for small values of $\tau$, suggesting that the tendency toward supervised learning should remain small. Otherwise, as in the case of $\tau = 0.5$, the model may quickly overfit to locally high-reward actions. Another configuration that will lead to degraded performance is when $t_3$ is set too large, such as $t_3 = 400$, where the model may fail to balance between exploration and exploitation.

Finally, we summarize the results of cPPO, cDPO, and hPPO with standard deviations. For each method and its corresponding baseline, we used the best-performing hyperparameters and conducted five runs with different random seeds. The improvements are both notable and stable, highlighting the potential for further research in this direction.

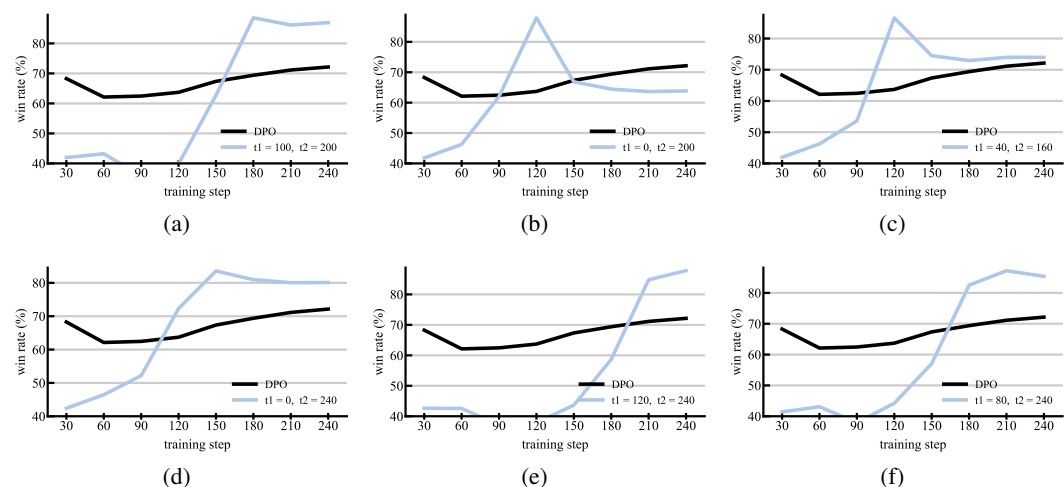

Figure 13: *DPO vs. cDPO*. For the Qwen3-1.7B model trained on UltraChat 200K and tested on HH-RLHF-helpfulness, we compare DPO with cDPO across hyper-parameters.

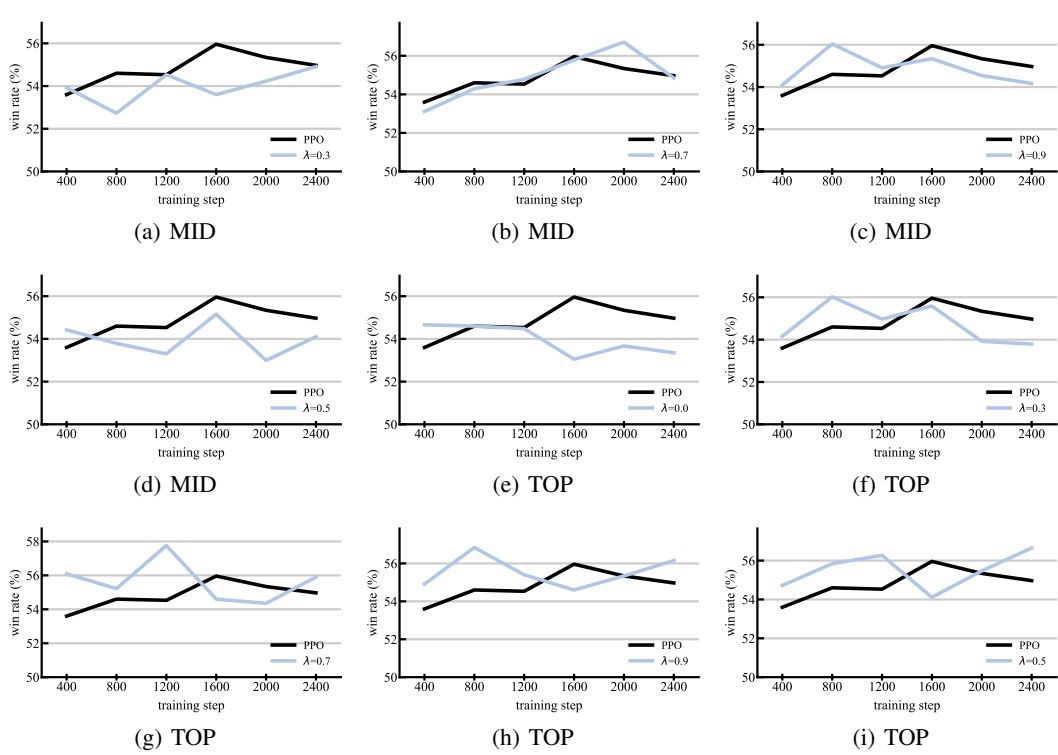

Figure 14: *PPO vs. cPPO*. For the Pythia-2.8B model trained on UltraChat 200K and tested on HH-RLHF-helpfulness, we compare PPO with cPPO across hyper-parameters. Both variants of cPPO are considered: one controlling the top weighted data (TOP) and the other controlling the middle weighted data (MID).

## E    THEORETICAL INSIGHTS

Although our exploration is not theory-driven, we still aim to provide some formal derivations to make it more rigorous. First, since we report the gradient alignment condition not at every PO step but at certain intervals, in Section E.1, we show that interval sampling is a good approximation to

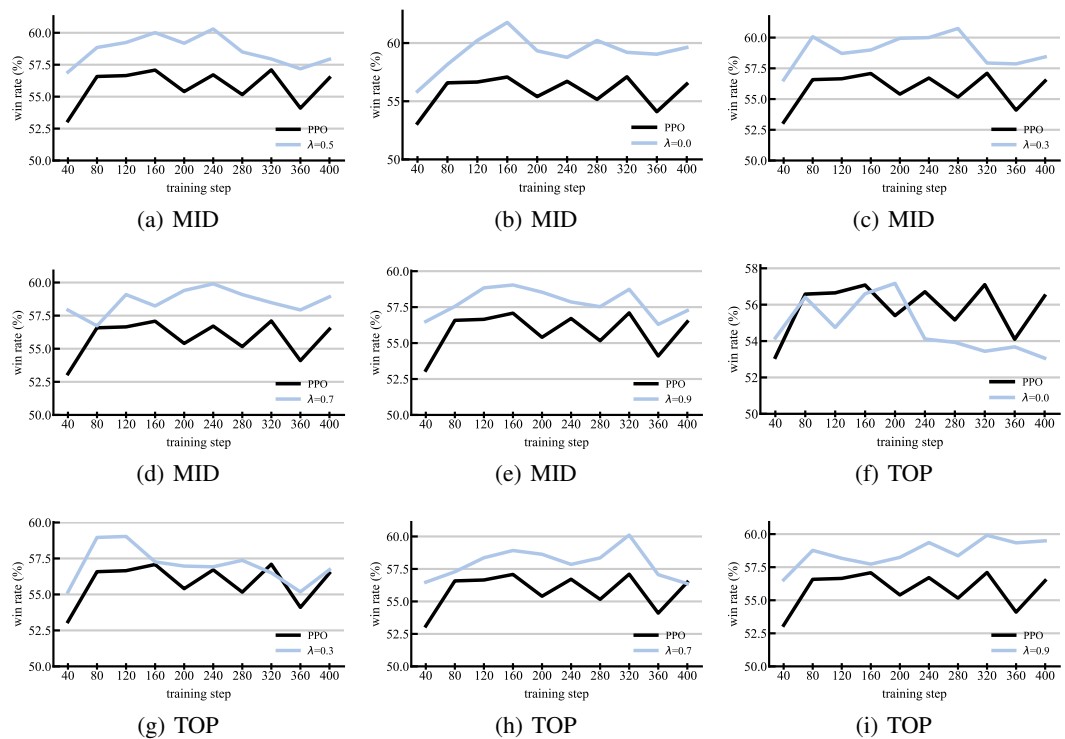

Figure 15: *PPO vs. cPPO*. For the Qwen3-1.7B model trained on UltraChat 200K and tested on HH-RLHF-helpfulness, we compare PPO with cPPO across hyper-parameters. Both variants of cPPO are considered: one controlling the top weighted data (TOP) and the other controlling the middle weighted data (MID).

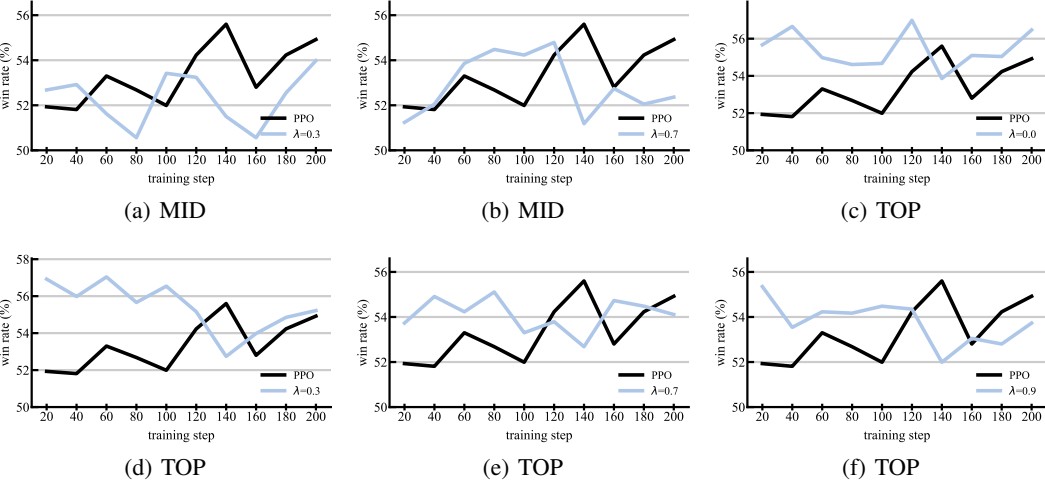

Figure 16: *PPO vs. cPPO*. For the Llama3-8B model trained on UltraChat 200K and tested on HH-RLHF-helpfulness, we compare PPO with cPPO across hyper-parameters. Both variants of cPPO are considered: one controlling the top weighted data (TOP) and the other controlling the middle weighted data (MID).

reporting at every PO step. Then, in Section E.2, we analyze the gradient flow for positive learning, negative learning, and loss reweighting, respectively. Note that we focus on the individual behaviors of these PO components, while leaving their mutual interactions to future work.

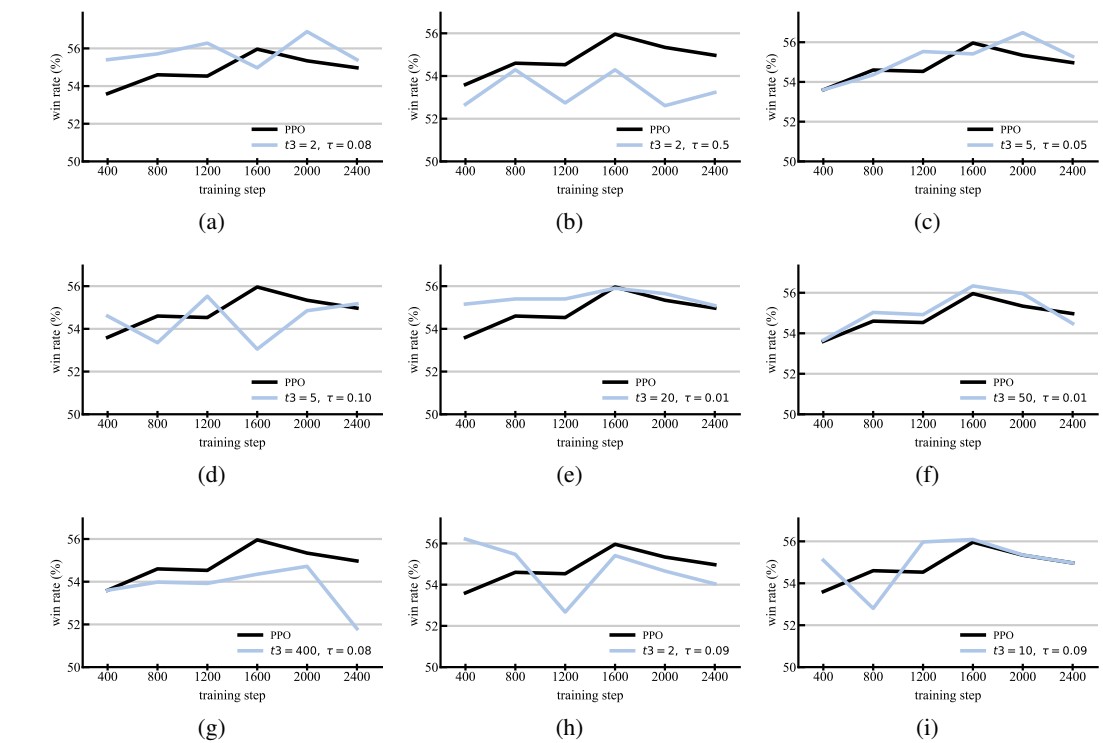

Figure 17: *PPO vs. hPPO*. For the Pythia-2.8B model trained on UltraChat 200K and tested on HH-RLHF-helpfulness, we compare PPO with hPPO across hyper-parameters.

Table 1: Win rates (mean $\pm$ standard deviation) of DPO and PPO, as well as their refined variants, i.e., cDPO, cPPO, and hPPO, across different models. Results are averaged over five runs with different random seeds using the best-performing hyperparameters.

| PO Method | Pythia-2.8B | Qwen3-1.7B | Llama3-8B |
|---|---|---|---|
| DPO | $76.71 \pm 0.20$ | $72.14 \pm 0.16$ | $67.71 \pm 0.27$ |
| cDPO | $\mathbf{79.88} \pm 0.13$ | $\mathbf{87.91} \pm 0.20$ | $\mathbf{69.54} \pm 0.06$ |
| PPO | $55.96 \pm 0.24$ | $57.10 \pm 0.18$ | $55.60 \pm 0.34$ |
| cPPO | $\mathbf{57.76} \pm 0.10$ | $\mathbf{61.77} \pm 0.15$ | $\mathbf{57.04} \pm 0.07$ |
| hPPO | $56.89 \pm 0.14$ | $61.17 \pm 0.12$ | $56.99 \pm 0.19$ |

## E.1 PREDICTIVE POWER OF GRADIENT ALIGNMENT

To see that reporting $\mathcal{G}$ at certain intervals is sufficient to capture the overall learning behaviors, we show that the current gradient alignment condition can indicate the trend in the near future. We consider stochastic mini-batch gradient updates for generality, where at each step $t$, we have a batch of data $\mathcal{D}_{(t)}$ randomly drawn from $\mathcal{D}$, and the parameter are updated from $\boldsymbol{\theta}_{(t)}$ to $\boldsymbol{\theta}_{(t+1)}$ as $\boldsymbol{\theta}_{(t+1)} = \boldsymbol{\theta}_{(t)} - \eta\nabla_{\boldsymbol{\theta}}\mathcal{L}(\mathcal{D}_{(t)};\boldsymbol{\theta}_{(t)})$. When further inspecting two consecutive steps, we have

$$\boldsymbol{\theta}_{(t+2)} = \boldsymbol{\theta}_{(t)} - \eta\nabla_{\boldsymbol{\theta}}\mathcal{L}\big(\mathcal{D}_{(t)};\boldsymbol{\theta}_{(t)}\big) - \eta\nabla_{\boldsymbol{\theta}}\mathcal{L}\big(\mathcal{D}_{(t+1)};\boldsymbol{\theta}_{(t)} - \eta\nabla_{\boldsymbol{\theta}}\mathcal{L}(\mathcal{D}_{(t)};\boldsymbol{\theta}_{(t)})\big), \qquad (21)$$

which can be further approximated by a first-order Taylor expansion around $\boldsymbol{\theta}_{(t)}$, following

$$\boldsymbol{\theta}_{(t+2)} \approx \boldsymbol{\theta}_{(t)} - \eta\big[\nabla_{\boldsymbol{\theta}}\mathcal{L}(\mathcal{D}_{(t)};\boldsymbol{\theta}_{(t)}) + \nabla_{\boldsymbol{\theta}}\mathcal{L}(\mathcal{D}_{(t+1)};\boldsymbol{\theta}_{(t)})$$
$$+ \nabla_{\boldsymbol{\theta}}^2\mathcal{L}(\mathcal{D}_{(t+1)};\boldsymbol{\theta}_{(t)})(-\eta\nabla_{\boldsymbol{\theta}}\mathcal{L}(\mathcal{D}_{(t+1)};\boldsymbol{\theta}_{(t)}))\big]. \qquad (22)$$

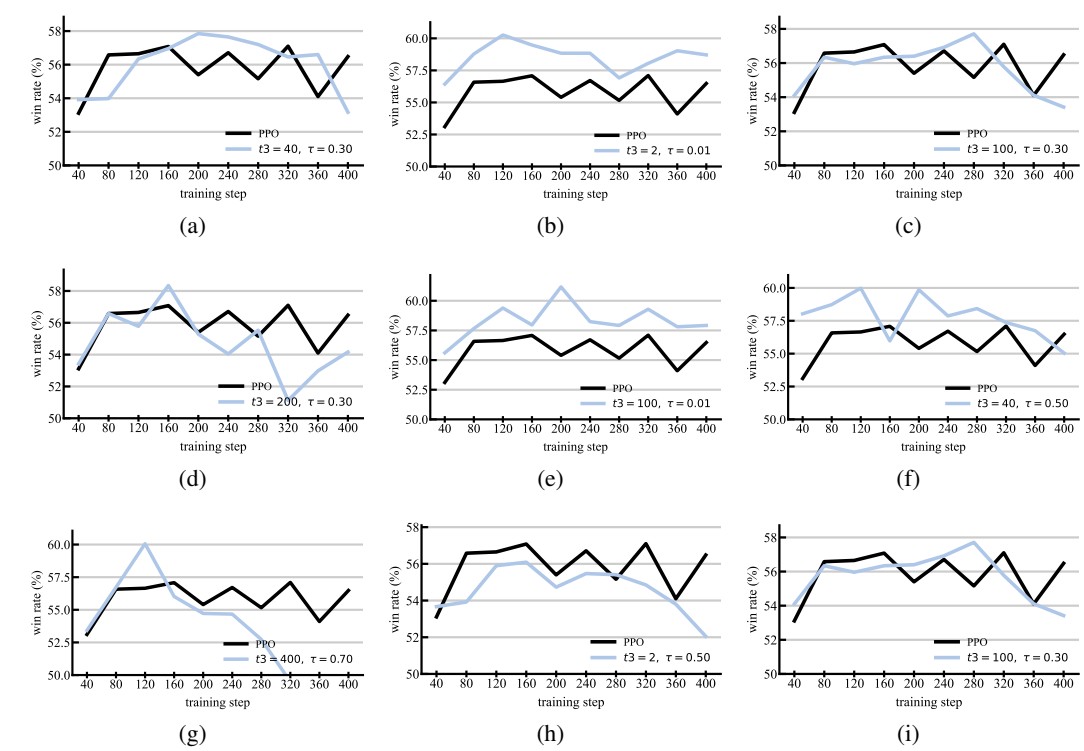

Figure 18: *PPO vs. hPPO*. For the Qwen3-1.7B model trained on UltraChat 200K and tested on HH-RLHF-helpfulness, we compare PPO with hPPO across hyper-parameters.

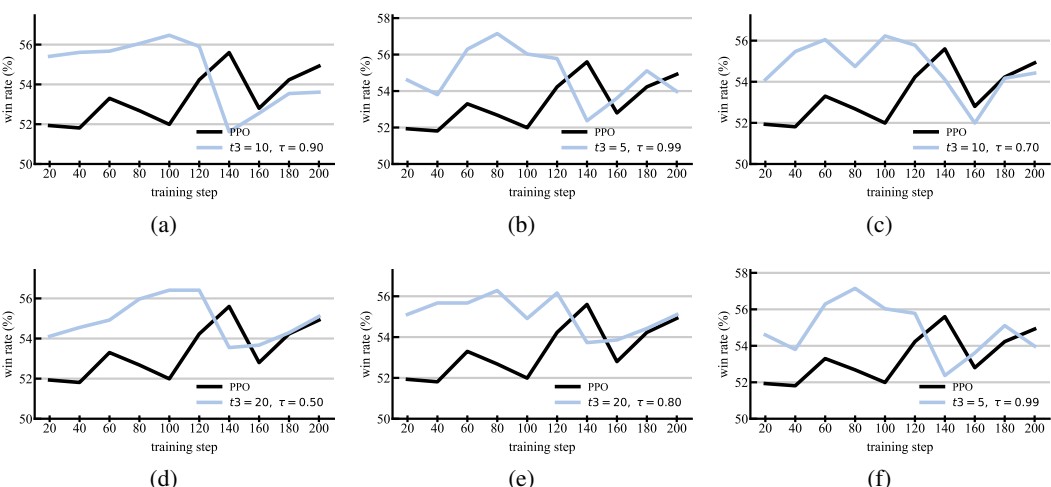

Figure 19: *PPO vs. hPPO*. For the Llama3-8B model trained on UltraChat 200K and tested on HH-RLHF-helpfulness, we compare PPO with hPPO across hyper-parameters.

Equation 22 as above can be further expanded to incorporate more updating steps. Considering the accumulated gradient updating from step $0$ to $T$, we have

$$\boldsymbol{\theta}_{(T)} \approx \boldsymbol{\theta}_{(0)} - \eta \sum_{t=0}^{T-1} \nabla_{\boldsymbol{\theta}} \mathcal{L}(\mathcal{D}_{(t)}; \boldsymbol{\theta}_{(0)}) + \sum_{t=1}^{T-1} \psi_{(t)} \tag{23}$$

where $\psi_{(t)} = -\eta \nabla_{\boldsymbol{\theta}}^2 \mathcal{L}(\mathcal{D}_{(t)}; \boldsymbol{\theta}_{(0)})\left(-\eta \sum_{t'=0}^{T-1} \nabla_{\boldsymbol{\theta}} \mathcal{D}(\mathcal{D}_{(t')}; \boldsymbol{\theta}_{(0)}) + \sum_{t'=0}^{T-1} \psi_{(t')}\right)$ and $\psi_{(0)} = 0$. Then, omitting higher-order terms due to small $\eta$, we can write

$$\boldsymbol{\theta}_{(T)} \approx \boldsymbol{\theta}_{(0)} - \eta E A \nabla_{\boldsymbol{\theta}} \mathcal{L}_{\mathrm{u}}(\mathcal{D}_{\mathrm{u}}; \boldsymbol{\theta}^{(0)}) \tag{24}$$

with $A = I - \mathtt{lr} \sum_{t=1}^{T-1} \nabla_{\boldsymbol{\theta}}^2 \mathcal{L}(S^{(t)}; \boldsymbol{\theta}^{(0)})$, $I$ the identify matrix, and $E$ the number of PO epochs. Regarding the performance measured by the negative log-likelihood, we again apply a first-order Taylor expansion around $\boldsymbol{\theta}_{(0)}$, and thus have

$$\begin{aligned}
\hat{\mathbb{E}}_{\mathcal{D}'}\big[-\log \pi(y'|x'; \boldsymbol{\theta}_{(T)})\big] &- \hat{\mathbb{E}}_{\mathcal{D}'}\big[-\log \pi(y'|x'; \boldsymbol{\theta}_{(0)})\big] \\
&\approx -\eta E \hat{\mathbb{E}}_{\mathcal{D}'}\big[-\nabla_{\boldsymbol{\theta}} \log \pi(y'|x'; \boldsymbol{\theta}_{(t)})\big]^{\top} A \nabla_{\boldsymbol{\theta}} \mathcal{L}(\mathcal{D}; \boldsymbol{\theta}_{(t)}).
\end{aligned} \tag{25}$$

Since $A$ is a symmetric and positive-definite matrix when $\eta$ is small, there exists a $\sigma \in [\sigma_1, \sigma_2]$, where $\sigma_1$ and $\sigma_2$ are the minimal and maximal eigenvalues of $A$, such that

$$\hat{\mathbb{E}}_{\mathcal{D}'}\big[-\nabla_{\boldsymbol{\theta}} \log \pi(y'|x'; \boldsymbol{\theta}_{(t)})\big]^{\top} A \nabla_{\boldsymbol{\theta}} \mathcal{L}(\mathcal{D}; \boldsymbol{\theta}_{(t)}) \approx \sigma \hat{\mathbb{E}}_{\mathcal{D}'}\big[-\nabla_{\boldsymbol{\theta}} \log \pi(y'|x'; \boldsymbol{\theta}_{(t)})\big]^{\top} \nabla_{\boldsymbol{\theta}} \mathcal{L}(\mathcal{D}; \boldsymbol{\theta}_{(t)}). \tag{26}$$

Substituting equation 26 back into equation 25, we obtain the same gradient alignment condition as in equation 3. Therefore, for sufficiently small $\eta$ (as is often the case in practical PO training), the gradient alignment condition can indicate the trend over a sequence of update steps. Consequently, monitoring $\mathcal{G}$ at intervals during PO training offers a reliable yet more computationally efficient way to analyze PO behavior. The derivations above are inspired by (Thudi et al., 2022).

### E.2 Individual Behaviors of PO Components

We further demonstrate the individual effects of positive learning, negative learning, and loss reweighting. First, we illustrate how positive and negative learning contribute to target shaping. Then, we conduct an examination of how loss reweighting facilitates distribution matching. Finally, we characterize the learning stability of these components.

**Target Shaping**. We consider a parametric conditional model $\pi(y|x; \boldsymbol{\theta})$ for the conditional distribution of $y$ given $x$ and an observed data distribution $p(x, y)$. Assume that there exists $\boldsymbol{\theta}^*$ such that $\pi(y|x; \boldsymbol{\theta}^*) = p(y|x)$, we approximate $\pi(y|x; \boldsymbol{\theta})$ around $\boldsymbol{\theta}^*$ to have $\pi(y|x; \boldsymbol{\theta}) = \pi(y|x; \boldsymbol{\theta}^*) + \nabla_{\boldsymbol{\theta}} \pi(y|x; \boldsymbol{\theta}^*)^{\top} \Delta\boldsymbol{\theta} + O(||\Delta\boldsymbol{\theta}||^2)$. Moreover, since $\nabla_{\boldsymbol{\theta}} \pi(y|x; \boldsymbol{\theta}^*) = \pi(y|x; \boldsymbol{\theta}^*) \nabla_{\boldsymbol{\theta}} \log \pi(y|x; \boldsymbol{\theta}^*) = p(y|x) \nabla_{\boldsymbol{\theta}} \log \pi(y|x; \boldsymbol{\theta}^*)$, we have

$$\pi(y|x; \boldsymbol{\theta}) \approx p(y|x) + p(y|x) \nabla_{\boldsymbol{\theta}} \log \pi(y|x; \boldsymbol{\theta}^*)^{\top} \Delta\boldsymbol{\theta}. \tag{27}$$

Approximating $\mathbb{E}_{p(x,y)}\big[\nabla_{\boldsymbol{\theta}} \log \pi(y|x; \boldsymbol{\theta})\big]$ around $\boldsymbol{\theta}^*$, we have $\mathbb{E}_{p(x,y)}\big[\nabla_{\boldsymbol{\theta}} \log \pi(y|x; \boldsymbol{\theta})\big] = -F\Delta\boldsymbol{\theta}$, where $F = -\mathbb{E}_{p(x,y)}\big[\nabla_{\boldsymbol{\theta}}^2 \log \pi(y|x; \boldsymbol{\theta}^*)\big]$ is the Fisher information matrix. Therefore,

$$\frac{d\Delta\boldsymbol{\theta}}{dt} = \frac{d\boldsymbol{\theta}}{dt} \approx -F\Delta\boldsymbol{\theta}. \tag{28}$$

Differentiating equation 27 w.r.t. $t$ and then combining with equation 28, we have

$$\frac{d}{dt} \pi(y|x; \boldsymbol{\theta}) \approx p(y|x) \nabla_{\boldsymbol{\theta}} \log \pi(y|x; \boldsymbol{\theta}^*)^{\top} \frac{d\Delta\boldsymbol{\theta}}{dt}, \tag{29}$$

$$\approx p(y|x) \nabla_{\boldsymbol{\theta}} \log \pi(y|x; \boldsymbol{\theta}^*)^{\top} (-F\Delta\boldsymbol{\theta}). \tag{30}$$

From equation 27, we also have $\nabla_{\boldsymbol{\theta}} \log \pi(y|x; \boldsymbol{\theta}^*)^{\top} \Delta\boldsymbol{\theta} \approx \frac{\pi(y|x; \boldsymbol{\theta}) - p(y|x)}{p(y|x)}$. If we further approximate $F$ as a positive, linear operator $c(x)$, we have

$$\frac{d}{dt} \pi(y|x; \boldsymbol{\theta}) \approx c(x)\big(p(y|x) - \pi(y|x; \boldsymbol{\theta})\big). \tag{31}$$

Therefore, we will have $\frac{d}{dt} \pi(y|x; \boldsymbol{\theta}) \approx c(x)\big(p(y|x) - \pi(y|x; \boldsymbol{\theta})\big)$ for positive learning, where $\pi(y|x; \boldsymbol{\theta})$ moves toward $p(y|x)$ as the target. Conversely, we can similarly derive $\frac{d}{dt} \pi(y|x; \boldsymbol{\theta}) \approx -c(x)\big(p(y|x) - \pi(y|x; \boldsymbol{\theta})\big)$ for negative learning, in which case $\pi(y|x; \boldsymbol{\theta})$ deviates from $p(y|x)$.

**Distribution Matching**. Consider the reweighted objective as $\mathbb{E}_{p(x,y)}\big[\omega(x, y) \log \pi(y|x; \boldsymbol{\theta})\big]$, we can rewrite it as an expectation under a new implicit distribution $q(x, y) \propto \omega(x, y) p(x, y)$, such that

$$\mathbb{E}_{p(x,y)}\big[\omega(x, y) \log \pi(y|x; \boldsymbol{\theta})\big] = \mathbb{E}_{q(x,y)}\big[\log \pi(y|x; \boldsymbol{\theta})\big]. \tag{32}$$

When minimizing equation 32, the derivation parallels that of the gradient flow for $\mathbb{E}_{p(x,y)}\big[\log \pi(y|x;\boldsymbol{\theta})\big]$. Therefore, we can write

$$\frac{d}{dt}\pi(y|x;\boldsymbol{\theta}) \approx c(x)\big(q(y|x) - \pi(y|x;\boldsymbol{\theta})\big), \tag{33}$$

which drives the prediction of $\pi$ to approach the implicit target distribution indicated by $q$.

**Learning Stability**. Upon stability, we conduct the small-perturbation analysis. For positive learning, we add a small perturbation $\epsilon(y|x)$ to the optimal $\pi(y|x;\boldsymbol{\theta}^*)$ and then linearize the equation. As $\frac{d}{dt}\pi(y|x;\boldsymbol{\theta}^*) = 0$, $\pi(y|x;\boldsymbol{\theta}^*) = p(y|x)$. Adding $\epsilon$ to $\frac{d}{dt}\pi(y|x;\boldsymbol{\theta})$ following equation 31, we have

$$\frac{d}{dt}\big(\pi(y|x;\boldsymbol{\theta}) + \epsilon(y|x)\big) = -\eta c(x)\epsilon(y|x), \tag{34}$$

and thus we derive $\dot{\epsilon}(y|x) = -\eta c(x)\epsilon(y|x)$. Solving this linear ODE function, we have

$$\epsilon(t) = \epsilon(0)e^{-\eta c(x)t} \tag{35}$$

for positive learning. Similarly, we have $\epsilon(t) = \epsilon(0)e^{\eta c(x)t}$ for negative learning. Since $c(x) > 0$ and $\eta > 0$, we have $\epsilon(0)e^{-\eta c(x)t} \to 0$ as $t \to \infty$, indicating stability for positive learning. In contrast, $\epsilon(0)e^{-\eta c(x)t} \to \infty$ as $t \to \infty$, indicating instability for negative learning. The same derivation holds for loss reweighting as in equation 35, as equation 35 is irrelevant to the data distribution.

## F LLM USAGE STATEMENT

In this paper, we utilized the commercial large language model GPT-5.1-Chat for language refinement and manuscript polishing. LLMs were not used for generating research ideas, designing methods, or conducting literature search and discovery.

