# OpenReview forum: "What Is Preference Optimization Doing, How and Why?"
_ICLR.cc/2026/Conference — Submitted to ICLR 2026_

### Official Review · Reviewer_i8HD · 2025-10-18

**Soundness:** 3
**Presentation:** 3
**Contribution:** 3
**Rating:** 6
**Confidence:** 3

**Summary:**

This paper investigates the training dynamics of preference optimisation when using a DPO and a PPO approach. To this end, they analyse the gradient alignment condition and look in more detail at the effect of the positive components, the negative components individually, as well as the impact of the gradient weight split into three tertiles. Moreover, they conduct ablation studies of dynamically adapting the positive and negative components, as well as the weighting, during preference optimisation training.

This paper gives a novel insight into the training dynamics of what is happening during PO in more detail, proposes some interesting hypotheses of the observed effects, and finally makes some initial proposals on how to improve the performance/gradient alignment when doing PO.

**Strengths:**

In my opinion, this paper introduces some new and novel insights into what is actually happening during preference optimisation, with a particular focus on the positive and negative components during training. I found their insight that DPO seems to overfit on the positive components and later focuses on the negative components, particularly interesting. I also appreciate their follow-up experiments in the appendix, which further validate their hypothesis.

In general, I find the paper mostly well written, and it's clear to follow. The experiments that are provided make sense to me and support the claims that the authors are making.

**Weaknesses:**

I think there are specific ways in which the paper could be further improved upon:
- The experiments lack any confidence interval, standard error, or at least standard deviation, which could indicate that the results are actually statistically significant. Especially in the win-rate experiments, reported in Figures 4b, d, e, and f (basically all the PPO ablations), the values of the win-rates seem to be within $\pm2$ pp, and I have a suspicion that they may not be statistically significant. Naturally, I understand that running the experiment multiple times is computationally expensive; therefore, the authors could consider, for example, bootstrapped confidence intervals of the test set.
- slightly related to the previous weakness, the proposed improvements on DPO and PPO, displayed in section 4, do not really seem to make a strong (or any) improvement over normal DPO and PPO. While we gain valuable insight into the dynamics of learning, the proposed solutions based on these insights do not seem particularly compelling. (Yet I understand that this is only a subpart of the paper, and the primary focus is on the insights.)
- I understand that the authors focus on one single model to demonstrate all the findings of the PO dynamics. I wonder how much these results are a result of the backbone model itself? Have you tested these experiments on a slightly newer backbone architecture, of similar size (e.g. Qwen25-3B, Llama3.2-3B, Gemma3-4B), and are the results still consistent?
- In Figure 4, to me it is not immediately clear what Cases 1,2,3 are, even after trying to find them in the text. Maybe the legend can be renamed
- Figures 1c and 2c are hard to distinguish between the different shades of green, and the curves often overlap.

**Questions:**

- Why are the win-rates of the DPO so much higher than the PPO approach? I know this is not directly linked to the insights of your paper, but I feel like they should be in the same range to make comparable claims that apply to both.
- What are cases 1,2, and 3 in Figure 4?
- Are the results actually statistically significant?
- Are the results consistent when using different LLM backbones?

---

> ### Author Response · Authors · 2025-11-21
>
> We sincerely thank you for your constructive comments and generous support, which mean a great deal to us. Our responses are provided below, and we hope they can sufficiently address your concerns.
>
> > **Q1**. The experiments lack any confidence interval, standard error, or at least standard deviation, which could indicate that the results are actually statistically significant. Especially in the win-rate experiments, reported in Figures 4b, d, e, and f (basically all the PPO ablations), the values of the win-rates seem to be within $\pm2$pp, and I have a suspicion that they may not be statistically significant. Naturally, I understand that running the experiment multiple times is computationally expensive; therefore, the authors could consider, for example, bootstrapped confidence intervals of the test set.
>
> **A1**. Following your suggestion, we report the results of cPPO, cDPO, and hPPO with standard deviations. For each method and its corresponding baseline, we used the best-performing hyperparameters and conducted five runs with different random seeds. Moreover, in addition to Pythia-2.8B, we also evaluated on Qwen3-1.7B and Llama-3-8B. The results are as follows.
>
> **Pythia-2.8B**
> |PO Method|Win Rate (%) ↑ (mean ± std)|
> |------|--------------|
> | DPO  |76.71 ± 0.20|
> | cDPO |**79.88** ± 0.13|
> | PPO  |55.96 ± 0.24|
> | cPPO |**57.76** ± 0.10|
> | hPPO |56.89 ± 0.14|
>
> **Qwen3-1.7B**
> | PO Method | Win Rate (%) ↑ (mean ± std) |
> |-----|----|
> | DPO | 72.14 ± 0.16|
> | cDPO| **87.91** ± 0.20|
> | PPO | 57.10 ± 0.18|
> | cPPO| **61.77** ± 0.15|
> | hPPO| 61.17 ± 0.12|
>
>
> **Llama-3-8B**
> |PO Method|Win Rate (%) ↑ (mean ± std))|
> |------|----|
> | DPO  | 67.71 ± 0.27|
> | cDPO | **69.54** ± 0.06|
> | PPO  | 55.60 ± 0.34|
> | cPPO | **57.04** ± 0.07|
> | hPPO | 56.99 ± 0.19|
>
>
> > **Q2**. Slightly related to the previous weakness, the proposed improvements on DPO and PPO, displayed in section 4, do not really seem to make a strong (or any) improvement over normal DPO and PPO. While we gain valuable insight into the dynamics of learning, the proposed solutions based on these insights do not seem particularly compelling. (Yet I understand that this is only a subpart of the paper, and the primary focus is on the insights.)
>
> **A2**. We apologize for the confusion. **The purpose of Figure 4 is to illustrate the impact of controlling sub-components of the original DPO and PPO objectives on performance curves**. Some results, such as Case 1 in Figure 4(d), may correspond to extreme hyperparameter settings that would not be used in practice. On the other hand, Figure 4 demonstrates that even without extensive hyperparameter tuning, **there exist many cases that outperform the original DPO and PPO (further verified in Appendix E), hightlighting the potential of behavior control**. We will include this clarification into our revision. Many thanks for pointing out this concern.

---

> ### Author Response · Authors · 2025-11-21
>
> > **Q3**. I understand that the authors focus on one single model to demonstrate all the findings of the PO dynamics. I wonder how much these results are a result of the backbone model itself? Have you tested these experiments on a slightly newer backbone architecture, of similar size (e.g. Qwen25-3B, Llama3.2-3B, Gemma3-4B), and are the results still consistent?
>
> **A3**. We used Pythia as the base model because it is open-sourced with detailed documentation of its training and fine-tuning procedures, making it well-suited for rigorous verification. For instance, we know that this base model has not been trained with any preference optimization procedures, thereby eliminating potential confounding factors. Few base models are this transparent, which is why we only adopted Pythia in our experiments.
>
> On the other hand, we fully agree that conducting experiments on a broader range of models would further strengthen the generality of our conclusions. Therefore, following your kind suggestion, **we additionally tested the gradient alignment conditions on Qwen3-1.7B**, with the results summarized as follows. As observed, **the conclusions drawn from Pythia also explain the learning dynamics in Qwen, demonstrating that our findings are general and reliable**. We will include the alignment-condition curves in our revised version to provide clearer illustrations.
>
>
> **DPO — Gradient Alignment**
>
> | Training Step | NEG (Negative) | POS (Positive) |TOT (Total)|
> |------|--------|----------------|----------------|
> | 30 | -1.948e+04| 2.848e+04|9.038e+03|
> | 60 | -4.267e+03| 1.032e+04|6.056e+03|
> | 90 | -4.430e+02| 5.345e+03|4.895e+03|
> | 120| -6.467e+01| 6.434e+03|6.368e+03|
> | 150| 7.253e+03| -5.907e+03|1.344e+03|
> | 180| 6.096e+03| -4.036e+03|2.063e+03|
> | 210| 6.372e+03| -5.467e+03|9.086e+02|
> | 240| 6.426e+03| -5.435e+03|1.002e+03|
>
>
> | Training Step | TOP | MID |BOT|
> |----|----------|-----|---------|
> | 30 | 6.265e+03| 2.388e+03|3.745e+02|
> | 60 |3.233e+03| 2.250e+03|5.705e+02|
> | 90 | 2.672e+03| 1.847e+03|3.759e+02|
> | 120| 4.647e+03| 1.361e+03|3.580e+02|
> | 150| 2.201e+01| 1.045e+03|3.185e+02|
> | 180| 3.025e+02| 1.471e+03|2.862e+02|
> | 210| -4.842e+02|1.098e+03|2.915e+02|
> | 240| -3.968e+02|9.858e+02|3.385e+02|
>
>
> **PPO — Gradient Alignment**
>
> | Training Step | NEG (Negative) | POS (Positive) |TOT (Total)|
> |------|--------|----------------|----------------|
> | 40| -7.303e+00| 6.007e+00|-1.298e+00|
> | 80| -7.589e+00| 5.937e+00|-1.651e+00|
> |120| -7.925e+00| 6.783e+00|-1.140e+00|
> |160| -8.330e+00| 6.555e+00|-1.772e+00|
> |200| -8.500e+00| 6.617e+00|-1.882e+00|
> |240| -8.595e+00| 6.717e+00|-1.877e+00|
> |280| -8.491e+00| 6.919e+00|-1.569e+00|
> |320| -8.625e+00| 7.083e+00|-1.543e+00|
> |360| -8.247e+00| 6.974e+00|-1.268e+00|
> |400| -8.227e+00| 6.648e+00|-1.574e+00|
>
> | Training Step | TOP | MID |BOT|
> |---------------|-----|-----|---|
> | 40 | -1.234e+00|-2.800e-01|2.026e-02|
> | 80 | -8.211e-01| -2.002e-01|4.441e-03|
> | 120| -5.304e-01| -3.095e-01|5.317e-03|
> | 160| -7.414e-01| -2.437e-01|-2.712e-03|
> | 200| -1.033e+00| -2.669e-01|-4.364e-02|
> | 240| -3.575e-01| -1.339e-01|-5.020e-02|
> | 280| -2.365e-01|-2.141e-02|2.325e-03|
> | 320| -2.321e-01|-2.975e-01|-6.778e-02|
> | 360| -9.123e-01|-1.090e-01|-2.179e-02|
> | 400| -2.363e-01|-6.816e-02|-2.841e-02
>
>
>
>
> > **Q4**. In Figure 4, to me it is not immediately clear what Cases 1,2,3 are, even after trying to find them in the text. Maybe the legend can be renamed.
>
> **A4**. We apologize for the confusion. In Figure 4, **each case corresponds to the win‑rate curve under a specific hyperparameter configuration**, with detailed and extended results provided in Appendix E in our paper. We agree that referring to these configurations simply as “Case 1–3” may cause misunderstanding, and we will revise the presentation to clearly indicate the associated hyperparameter settings.
>
>
> > **Q5**. Figures 1c and 2c are hard to distinguish between the different shades of green, and the curves often overlap.
>
> **A5**. We will address this clarity issue in our revision. Many thanks for pointing out this concern.

---

> ### Author Response · Authors · 2025-11-21
>
> > **Q6**. Why are the win-rates of the DPO so much higher than the PPO approach? I know this is not directly linked to the insights of your paper, but I feel like they should be in the same range to make comparable claims that apply to both.
>
> **A6**. Several factors may lead to this observation. First, the performance of PPO depends heavily on the reward model, whereas DPO relies more directly on the preference dataset. Since these two sources offer different types of learning signals and diverse information, DPO and PPO are not strictly comparable. **Although we follow the common practice, the adopted reward model might not be our optimal choice, which could affect the PPO results**.
>
> Moreover, online PO methods such as PPO are generally more sensitive to the base model, as their iterative rollouts can amplify errors when the initial policy is weak. In contrast, DPO, being an offline method, tends to be more stable when the base model is less capable. Hence, **the lower win rate of PPO in our experiments could also result from the inherent issues of the base model**.
>
> In practice, industrial LLM pipelines face similar scenarios. This is why practioners often first apply DPO to align preference data and then use PPO (or other online RL methods) to further refine model behaviors. However, as a research paper, **we tend to follow the academic practice that evaluates DPO and PPO independently**. We have tried our best to tune their hyperparameters such that each method could be tested under its best achievable configuration.

---

> ### Author Response · Authors · 2025-11-25
>
> Dear Reviewer i8HD,
>
> We sincerely thank you for your efforts in reviewing our work and for your great support! Please let us know if you need any further information or if there are additional points you would like to discuss with us. We would be glad to answer any questions of your interest.
>
> Thank you very much!
>
> Best regards,
>
> Authors of #4235

---

> ### Comment · Reviewer_i8HD · 2025-11-25
>
> *A1)*
> Thank you very much for testing over multiple runs and reporting the standard deviation on the win-rate experiments.
> The results appear significant, especially for the DPO comparison.
>
> When you ran the multiple experiments, could you also include confidence intervals in the plots, as in figures 2, 3, and 4?
>
> *A2)*
> Thank you for pointing out question 2 on figure 4. I had a further look at Appendix Section E, and maybe I am just misunderstanding the experiments or figures. Still, to me, the statement: "there exist many cases that outperform the original DPO and PPO" seems a bit like an overclaim. There are some training step instances where the win rate of the controlled method outperforms DPO or PPO, but this is true in only a minority of cases. Also, again, without a CI bound around it, it doesn't seem easy to determine how much this is just stochasticity from the optimiser and data.
>
> Maybe you can clarify my misunderstandings.
>
> Also, I believe you can upload a new version of the manuscript, and that could potentially make the reviewing of the latest results a bit easier.

---

> ### Comment · Reviewer_i8HD · 2025-11-25
>
> *A3)*
> Thank you. I acknowledge the additional experiments conducted on Qwen3-7B and confirm that the results are consistent.
>
> *A4)*
> Thanks for clarifying. I agree with the authors and encourage adding the description of "Cases 1 to 3" in the main manuscript if it is referenced there
>
> *A5)*
> The clarification is acknowledged and appreciated.
>
> *A6)*
> The clarification is acknowledged and appreciated.

---

> > ### Author Response · Authors · 2025-11-27
> >
> > Sincere thanks for your feedback and generous support. We have incorporated your suggestions into our revision and will continue to refine the draft. For your further comments, please see our responses below.
> >
> > > **Q1**. When you ran the multiple experiments, could you also include confidence intervals in the plots, as in figures 2, 3, and 4?
> >
> > **A1**. Due to the high computational cost and limited time, we regret that we cannot provide confidence intervals for all these figures in the current revision. We fully agree that confidence intervals would strengthen our conclusions, and we plan to conduct the corresponding experiments carefully and include them in a future version.
> >
> > > **Q2**. Thank you for pointing out question 2 on figure 4. I had a further look at Appendix Section E, and maybe I am just misunderstanding the experiments or figures. Still, to me, the statement: "there exist many cases that outperform the original DPO and PPO" seems a bit like an overclaim. There are some training step instances where the win rate of the controlled method outperforms DPO or PPO, but this is true in only a minority of cases. Also, again, without a CI bound around it, it doesn't seem easy to determine how much this is just stochasticity from the optimiser and data.
> >
> > **A2**. We apologize that our previous explanation was still unclear. From the perspective of achieving improved performance, our reported results can be regarded as part of the hyperparameter tuning process. The final improvements, together with confidence intervals, are summarized in the tables below (same results as in A1 of our previous responses), which we have added to Appendix D. We have also refined the discussion in Section 4 so that readers can quickly refer to Appendix D.
> >
> >
> > **Pythia-2.8B**
> > |PO Method|Win Rate (%) ↑ (mean ± std)|
> > |------|--------------|
> > | DPO  |76.71 ± 0.20|
> > | cDPO |**79.88** ± 0.13|
> > | PPO  |55.96 ± 0.24|
> > | cPPO |**57.76** ± 0.10|
> > | hPPO |56.89 ± 0.14|
> >
> > **Qwen3-1.7B**
> > | PO Method | Win Rate (%) ↑ (mean ± std) |
> > |-----|----|
> > | DPO | 72.14 ± 0.16|
> > | cDPO| **87.91** ± 0.20|
> > | PPO | 57.10 ± 0.18|
> > | cPPO| **61.77** ± 0.15|
> > | hPPO| 61.17 ± 0.12|
> >
> >
> > **Llama3-8B**
> > |PO Method|Win Rate (%) ↑ (mean ± std))|
> > |------|----|
> > | DPO  | 67.71 ± 0.27|
> > | cDPO | **69.54** ± 0.06|
> > | PPO  | 55.60 ± 0.34|
> > | cPPO | **57.04** ± 0.07|
> > | hPPO | 56.99 ± 0.19|

---

### Official Review · Reviewer_CKPZ · 2025-10-21

**Soundness:** 1
**Presentation:** 1
**Contribution:** 2
**Rating:** 2
**Confidence:** 4

**Summary:**

This paper proposes a metric, *i.e.*, gradient alignment, to quantize the contribution of gradient descent to the log-probability of the final answer.
The optimization dynamics of two popular post-training algorithms, *i.e.*, DPO and PPO, are then analyzed.
The conclusions include

* DPO behaves like supervised fine-tuning as it has relatively stable targets
* As training progresses, negative learning dominates target shaping, while positive learning prevents collapse
* The implicit reward is not reliable but primarily serves as a regularizer to mitigate over-fitting
* PPO behaves like reinforcement learning as its exploration covers a broad range of conflicting responses
* Positive learning encourages discovery for new targets while negative learning fosters further exploration
* Loss re-weighting controls exploration

Multiple variants, *i.e.*, cDPO, cPPO, hPPO, are also proposed to ablate the effect of negative learning.

**Strengths:**

* The components of preference learning, *e.g.*, positive and negative learning and loss re-weighting, are analyzed thoroughly.
* Ablation study strengthens the persuasiveness of the conclusions and provides insights for future research.

**Weaknesses:**

I deem that several logical flaws hinders the soundness of the conclusions, so I lean to reject the paper.
I would like to raise my score if these concerns are well addressed.

* L105: Why does the distinction between SFT and RL lie in whether they have relatively stable targets?
I deem the difference between SFT and RL lies in whether they learn from demonstrations or rewards.
* L134 (Minor): I do not think the objective is inherently non-differentiable.
* L143: It is not very clear to me why the log-probability of final answer rather than the ground truth is considered.
L108 claimed that SFT is expected to steadily progress toward the targets, while the final answer is not necessarily the target.
* L157 (Minor): I deem the design of gradient alignment can be regarded as extension of [1], which may be cited and discussed.
* L160: It is claimed that the difference between SFT and RL lies in whether they have stable objectives, and here it is classified based on the value of gradient alignment.
I understand that positive gradient alignment indicates that the gradient descent increases the log-probability of the final answer.
Why does this also indicates a stable objective?
* L168: Only a single setting is performed so that it is not clear how well the conclusions can be generalized.
* L315 (Minor): I think $\hat{A}$ inherently can be negative without estimation and normalization.

[1] Estimating Training Data Influence by Tracing Gradient Descent, NeurIPS 2020.

**Questions:**

* L255: What is the unbiasedness of learning objective of DPO?
Why does that only hold at the optimal parameter?
* L285: Is there any evidence to support the proposal?
* L335: Is there any reference to support such definition for positive and negative learning and loss reweighting?

---

> ### Author Response · Authors · 2025-11-21
>
> We sincerely thank you for your constructive and detailed comments, which are invaluable for improving the quality of our paper. We hope that the following responses could address your concerns, and we always welcome any further discussions and suggestions.
>
> > **Q1**. Why does the distinction between SFT and RL lie in whether they have relatively stable targets? I deem the difference between SFT and RL lies in whether they learn from demonstrations or rewards.
>
> **A1**. We fully agree that SFT and RL can indeed be distinguished from the perspectives of rewards, and we acknowledge that the mainstream literature has devoted many effort to these viewpoints. However, **as in many other areas of research, the same phenomenon can often be examined from different perspectives**. Our goal is to provide such a new angle focusing on the learning dynamics in shaping targets, which we believe they are very important, as also kindly recognized by you and many other reviewers. Below, please allow me to quickly review the structure and key findings of our study, further highlighting our contributions.
>
> In Section 2, we distinguished between supervised and reinforcement learning from the perspective of learning dynamics. In Section 3, we offered extensive experiments, demonstrating that DPO is more closely aligned with supervised learning, whereas PPO is more aligned with reinforcement learning. Furthermore, **analyzing learning dynamics enables fine-grained, component-level analysis that prior studies seldom achieve**. For example, we showed that in DPO, positive and negative learning jointly shape the learning targets, whereas in PPO, positive learning shapes the targets while negative learning ensure exploration.
>
> **These observations motivated us to control the learning dynamics to improve baselines, as discussed in Section 3**. Accordingly, we proposed promising variants, such as cDPO, cPPO, and hPPO, that demonstrated notable improvements over their original versions. Furthermore, **we explained the succes of several recent works in Appendix B based on our analysis**, and **envisioned the opportunity for coordinated preference optimization in Section 5**, which may inspire further research.
>
> Based on these findings and contributions, **we overalll believe it is valuable to conduct an analytic study grounded in learning targets**, as presented in this work.
>
>
> > **Q2**. I do not think the objective is inherently non-differentiable.
>
>
> **A2**. We sincerely appreciate your detailed comments and thoughtful suggestions. However, regarding this particular concern, we hold a different opinion. Heuristically, the objective $\mathbb E_{(x,y)\sim\pi_\theta} R(x,y)$ represents the expected rewards obtained from samples generated by the policy $\pi_\theta$. **Because this formulation depends on sampled outcomes rather than directly on differentiable quantities such as likelihoods**, it cannot be optimized directly through gradient-based methods. This is the reason why the policy gradient trick is essential in reinforcement learning, as it transforms a non-differentiable optimization problem into a differentiabl equivalent. We will clarify this point more explicitly in our revision.

---

> > ### Comment · Reviewer_CKPZ · 2025-11-26
> >
> > > A1. We fully agree that SFT and RL can indeed be distinguished from the perspectives of rewards, and we acknowledge that the mainstream literature has devoted many effort to these viewpoints. However, as in many other areas of research, the same phenomenon can often be examined from different perspectives. Our goal is to provide such a new angle focusing on the learning dynamics in shaping targets, which we believe they are very important, as also kindly recognized by you and many other reviewers. Below, please allow me to quickly review the structure and key findings of our study, further highlighting our contributions.
> >
> > This concern is not with the contribution, but with the justification of the claim.
> > The authors state that SFT has a stable target while RL does not.
> > Is this based on prior work or from experiments of this paper?
> > If this is merely the authors' personal opinion, it is insufficient.
> >
> > > A2. We sincerely appreciate your detailed comments and thoughtful suggestions. However, regarding this particular concern, we hold a different opinion. Heuristically, the objective $\mathbb E_{(x,y)\sim\pi_\theta} R(x,y)$ represents the expected rewards obtained from samples generated by the policy $\pi_\theta$. Because this formulation depends on sampled outcomes rather than directly on differentiable quantities such as likelihoods, it cannot be optimized directly through gradient-based methods. This is the reason why the policy gradient trick is essential in reinforcement learning, as it transforms a non-differentiable optimization problem into a differentiabl equivalent. We will clarify this point more explicitly in our revision.
> >
> > I regard that the following equation holds mathematically
> >
> > $\nabla_\theta \mathbb E_{y\sim \pi_\theta(\cdot|x)} [R(x,y)]=\mathbb E_{y\sim\pi_\theta(\cdot|x)}[R(x,y)\nabla_\theta \log\pi_\theta(y|x)]$
> >
> > Please note that this is **expectation** rather than **mean of Monte-Carlo samples**.
> > Mean of Monte-Carlo samples is a common estimator of expectation.
> > The expectation is differentiable while mean of Monte-Carlo samples is not.
> > The authors appear to be confusing the two concepts.

---

> ### Author Response · Authors · 2025-11-21
>
> > **Q3**. It is not very clear to me why the log-probability of final answer rather than the ground truth is considered. L108 claimed that SFT is expected to steadily progress toward the targets, while the final answer is not necessarily the target.
>
> **A3**. Using the final answer rather than the ground truth is one of our key design choices in our analytical framework. We are so glad that you raised this queston and would like to explain it further as follows.
>
> First, **generative models differ from discriminative models, as the former do not have a discrete target space and well-defined ground truths**. This is why many PO methods rely on reward functions to assess whether the policy has improved, and why win rate, rather than accuracy, is typically used to evaluate performance (even in math reasoning, some correct anwers may be much preferred). Therefore, it is more appropriate for generative models to analyze how the learning dynamics evolve toward producing final responses.
>
> Second, even for offline methods such as DPO, **the preferred data do not necessarily represent the ground truth**, otherwise SFT on the preferred data alone would be enough. We have verified this phenomenon in Section 3.1, where we have showed that those dispreferred data can also contribute to target shaping. This suggests that **there exist some implicit learning targets that DPO relies on but to which we have no direct access**. A similar discussion holds for online methods like PPO, though it is conceptually easier to understand. Only the question part is adopted in PPO training, while **the reward function guides the policy in shaping its responses**. Therefore, there is likewise no explicit ground truth, and the reward function implicitly defines the directions for the underlying targets.
>
> Overall, **we believe that using final responses instead of ground truth offers a novel perspective within the literature**, enabling gradient-based analyses originally designed for discriminative models to generalize properly for generative models. We will provide additional explanation in our revision.
>
> > **Q4**. I deem the design of gradient alignment can be regarded as extension of [1], which may be cited and discussed.
>
> **A4**. Many thanks for your suggested reference. We fully agree that gradient-based analysis is a well-established and widely studied approach in prior work. Eq. 1 in [1] is exactly the formulation we adopt. Other studies, such as Eq. 2 in [2], can also be viewed as a form of gradient-based analysis, but further incorporating the Fisher information to capture the parameter geometry. However, our focus differs substantially from these previous works. We elaborate on these differences below.
>
> First, **we focus on generative models, whereas previous works mainly studied discriminative models**. As mentioned in A3, this shift introduces new challenges, as PO objectives may involve implicit targets and lack explicit ground truths. Second, previous works typically emphasize the data perspective in their analysis, but **we further explore the objective (cf., Section 3) and the model perspectives (cf., Appendix D)**. Our analysis uncovers some intriguing findings that are rarely discussed in existing literature, offering deeper insights and potentially inspiring future research, as outlined in A1. We will further clarify the novelty and significance of our work in comparison with previous studies in our revision. Thanks again for your kind suggestion.
>
> [2] Koh et al. Understanding Black-box Predictions via Influence Functions. In ICML, 2017.
>
> > **Q5**. It is claimed that the difference between SFT and RL lies in whether they have stable objectives, and here it is classified based on the value of gradient alignment. I understand that positive gradient alignment indicates that the gradient descent increases the log-probability of the final answer. Why does this also indicates a stable objective?
>
> **A5**. Sorry for the confusion. The stability we refer to is indicated by the latter part of the sentence, i.e., "for the majortiy of $t$". Intuitively, if only a few steps satisfy the positive gradient alignment condition, it suggests instability, and it would be hard to say the learning behavior as supervised learning. We will make this point clearer in our revision.

---

> ### Author Response · Authors · 2025-11-21
>
> > **Q6**. Only a single setting is performed so that it is not clear how well the conclusions can be generalized.
>
> **A6**. We used Pythia as the base model because it is open-sourced with detailed documentation of its training and fine-tuning procedures, making it well-suited for rigorous verification. For instance, we know that this base model has not been trained with any preference optimization procedures, thereby eliminating potential confounding factors. Few base models are this transparent, which is why we only adopted Pythia in our experiments.
>
> On the other hand, we fully agree that conducting experiments on a broader range of models would further strengthen the generality of our conclusions. Therefore, following your kind suggestion, **we additionally tested the gradient alignment conditions on Qwen3-1.7B**, with the results summarized as follows.
>
> **DPO — Gradient Alignment**
>
> | Training Step | NEG (Negative) | POS (Positive) |TOT (Total)|
> |------|--------|----------------|----------------|
> | 30 | -1.948e+04| 2.848e+04|9.038e+03|
> | 60 | -4.267e+03| 1.032e+04|6.056e+03|
> | 90 | -4.430e+02| 5.345e+03|4.895e+03|
> | 120| -6.467e+01| 6.434e+03|6.368e+03|
> | 150| 7.253e+03| -5.907e+03|1.344e+03|
> | 180| 6.096e+03| -4.036e+03|2.063e+03|
> | 210| 6.372e+03| -5.467e+03|9.086e+02|
> | 240| 6.426e+03| -5.435e+03|1.002e+03|
>
>
> | Training Step | TOP | MID |BOT|
> |----|----------|-----|---------|
> | 30 | 6.265e+03| 2.388e+03|3.745e+02|
> | 60 |3.233e+03| 2.250e+03|5.705e+02|
> | 90 | 2.672e+03| 1.847e+03|3.759e+02|
> | 120| 4.647e+03| 1.361e+03|3.580e+02|
> | 150| 2.201e+01| 1.045e+03|3.185e+02|
> | 180| 3.025e+02| 1.471e+03|2.862e+02|
> | 210| -4.842e+02|1.098e+03|2.915e+02|
> | 240| -3.968e+02|9.858e+02|3.385e+02|
>
>
> **PPO — Gradient Alignment**
>
> | Training Step | NEG (Negative) | POS (Positive) |TOT (Total)|
> |------|--------|----------------|----------------|
> | 40| -7.303e+00| 6.007e+00|-1.298e+00|
> | 80| -7.589e+00| 5.937e+00|-1.651e+00|
> |120| -7.925e+00| 6.783e+00|-1.140e+00|
> |160| -8.330e+00| 6.555e+00|-1.772e+00|
> |200| -8.500e+00| 6.617e+00|-1.882e+00|
> |240| -8.595e+00| 6.717e+00|-1.877e+00|
> |280| -8.491e+00| 6.919e+00|-1.569e+00|
> |320| -8.625e+00| 7.083e+00|-1.543e+00|
> |360| -8.247e+00| 6.974e+00|-1.268e+00|
> |400| -8.227e+00| 6.648e+00|-1.574e+00|
>
> | Training Step | TOP | MID |BOT|
> |---------------|-----|-----|---|
> | 40 | -1.234e+00|-2.800e-01|2.026e-02|
> | 80 | -8.211e-01| -2.002e-01|4.441e-03|
> | 120| -5.304e-01| -3.095e-01|5.317e-03|
> | 160| -7.414e-01| -2.437e-01|-2.712e-03|
> | 200| -1.033e+00| -2.669e-01|-4.364e-02|
> | 240| -3.575e-01| -1.339e-01|-5.020e-02|
> | 280| -2.365e-01|-2.141e-02|2.325e-03|
> | 320| -2.321e-01|-2.975e-01|-6.778e-02|
> | 360| -9.123e-01|-1.090e-01|-2.179e-02|
> | 400| -2.363e-01|-6.816e-02|-2.841e-02
>
>
>
> As observed, **the conclusions drawn from Pythia also explain the learning dynamics in Qwen, demonstrating that our findings are general and reliable**.
>
> Furthermore, **we also evaluate cPPO, cDPO, and hPPO on Qwen3-1.7B and Llama-3-8B**, comparing their win rates against the original PPO and DPO under their best-performing settings, as summarized below.
>
> **Pythia-2.8B**
> |PO Method|Win Rate (%) ↑ (mean ± std)|
> |------|--------------|
> | DPO  |76.71 ± 0.20|
> | cDPO |**79.88** ± 0.13|
> | PPO  |55.96 ± 0.24|
> | cPPO |**57.76** ± 0.10|
> | hPPO |56.89 ± 0.14|
>
> **Qwen3-1.7B**
> | PO Method | Win Rate (%) ↑ (mean ± std) |
> |-----|----|
> | DPO | 72.14 ± 0.16|
> | cDPO| **87.91** ± 0.20|
> | PPO | 57.10 ± 0.18|
> | cPPO| **61.77** ± 0.15|
> | hPPO| 61.17 ± 0.12|
>
>
> **Llama-3-8B**
> |PO Method|Win Rate (%) ↑ (mean ± std))|
> |------|----|
> | DPO  | 67.71 ± 0.27|
> | cDPO | **69.54** ± 0.06|
> | PPO  | 55.60 ± 0.34|
> | cPPO | **57.04** ± 0.07|
> | hPPO | 56.99 ± 0.19|
>
>
>
> We will include the performance and alignment-condition curves in our revised version to provide clearer illustrations.
>
>
> > **Q7**.  I think $\hat A$ inherently can be negative without estimation and normalization.
>
> **A7**. We agree that even with the original $A$, its values can still be negative. We will correct it in our revision and we sincerely appreciate your careful and detailed review.

---

> ### Author Response · Authors · 2025-11-21
>
> > **Q8**. What is the unbiasedness of learning objective of DPO? Why does that only hold at the optimal parameter?
>
> **A8**. The derivcation of DPO substitutes the **optimal policy** $\pi^*(y|x)\propto\pi_{\mathrm{ref}}(y|x)\exp(\frac{1}{\beta}r(x,y))$ back into the Bradley–Terry model, implicitly assuming that the policy being learned shares the same functional form as the optimal policy under some implicit reward. However, **this assumption creastes a circular dependence**, where the implicit reward $r(x,y;\theta)=\beta\log\frac{\pi(y|x;\theta)}{pi(y|x;\theta_{{\mathrm{ref}}})}$ depends on the policy $\pi(y|x;\theta)$ to be optimized based on this same reward.
>
> So, **if $\pi(y|x;\theta)$ is far from the optimal, the corresponding  $r(x,y;\theta)$ may become a biased estimate of the true latent reward**. When checking the theoretical guarantees for the original paper of DPO, we can find that the authors avoid proving convergence to the optimal $\pi^*$ or the monotonic improvement of the expected reward.
>
> > **Q9**. Is there any evidence to support the proposal?
>
> **A9**. A straightfoward way of justification is to examine the relationship between the implicit reward $\omega(x,y^+,y^-)$ and the preference margin $\log\pi(y^+|x;\theta)-\log\pi(y^-|x;\theta)$. It is easy to see that the implicit reward decreases as the margin increases. Generally, it implies that for well-learned data pairs, DPO assigns smaller training weights, thereby reducing its focus on those samples. This behavior will mitigate overfitting and thus serves as a form of regularization. We apologize for any confusion caused and will clarify this point further in our revision.
>
> > **Q10**. Is there any reference to support such definition for positive and negative learning and loss reweighting?
>
> **A10**. **Similar formulations of loss reweighting are widely applied across various areas of machine learning**, such as label noise learning [1] and meta-learning [2]. In these studies, the weights are typically assumed to be nonnegative. Otherwise, the convergence properties and the optimization stability can be adversely affected from a classical point of view. Distinguishing between positive and negative learning is relatively uncommon in traditional machine learning but has been discussed in the context of complementary label learning [3]. **However, this distinction has recently attracted increasing attention for LLMs**, particularly during post-training, where related ideas have emerged in areas such as LLM unlearning [4] and preference optimization [5]. We will add the related discussion in our revision.
>
>
> [1] Liu and Tao. Classification with Noisy Labels by Importance Reweighting. IEEE Transactions on Pattern Analysis and Machine Intelligence, 2015.
>
> [2] Ren et al. Learning to Reweight Examples for Robust Deep Learning. In ICML, 2018.
>
> [3] Ishida et al. Learning from Complementary Labels. In NIPS, 2017.
>
> [4] Zhang et al. Negative Preference Optimization: From Catastrophic Collapse to Effective Unlearning. In COLM, 2024.
>
> [5] Zhu et al. The Surprising Effectiveness of Negative Reinforcement in LLM Reasoning. Arxiv Preprint, 2025.

---

> > ### Comment · Reviewer_CKPZ · 2025-11-26
> >
> > > A8. When checking the theoretical guarantees for the original paper of DPO, we can find that the authors avoid proving convergence to the optimal $\pi^*$ or the monotonic improvement of the expected reward.
> >
> > I would like to clarify the theory foundation of DPO.
> > When $\pi$ is not optimal, the **induced reward** $r$ is indeed not the **true latent reward** $r^*$, but the policy $\pi$ is optimal under its induced reward *r*.
> > Given that any policy is optimal with respect to its induced reward, it only requires to perform reward modeling for the induced reward.
> > When the induced reward is (nearly) the true latent reward, the policy is (nearly) optimal (under the true latent reward).

---

> > > ### Author Response · Authors · 2025-11-27
> > >
> > > We are so glad to have such a senior reviewer like you, who not only offers so many insightful and detailed suggestions but also carefully reads our responses and provides further feedbacks. This means even more to us than receiving a positive score, and we always welcome further discussion.
> > >
> > > > **Q1**. This concern is not with the contribution, but with the justification of the claim. The authors state that SFT has a stable target while RL does not. Is this based on prior work or from experiments of this paper? If this is merely the authors' personal opinion, it is insufficient.
> > >
> > > **A1**. The distinction between supervised learning and reinforcement learning is well established in the literature, for example in Chapter 14 of [1] and Chapter 1 of [2]. Overall, in supervised learning, **the learner passively receives a labeled dataset, so the learning targets are explicit and relatively stable**. In contrast, reinforcement learning needs to balance exploration and exploitation: **exploitation aims to obtain high reward from already known actions (local targets), while exploration aims to discover better actions for the future (global targets). As a result, the effective targets in reinforcement learning are inherently dynamic**.
> > >
> > > We agree that adding explicit citations will strengthen our claim. We have already cited [1] in the original manuscript, and we will include [2] in our revision. We would also be happy to discuss this point further: if you are aware of any counterexamples where reinforcement learning has stable targets and supervised learning has changing targets under their classical definitions, we would be very interested to learn about them and refine our discussion accordingly.
> > >
> > > [1] Mohri, M., Rostamizadeh, A., & Talwalkar, A. (2018). Foundations of Machine Learning. MIT Press.
> > >
> > > [2] Sutton, R. S., & Barto, A. G. (2018). Reinforcement Learning: An Introduction. MIT Press.
> > >
> > > > **Q2**. Please note that this is expectation rather than mean of Monte-Carlo samples. Mean of Monte-Carlo samples is a common estimator of expectation. The expectation is differentiable while mean of Monte-Carlo samples is not. The authors appear to be confusing the two concepts.
> > >
> > > **A2**. We realize your point and agree with it. We will revise the corresponding description to make it more rigorous. Thank you for your detailed clarification and helpful guidance. We sincerely appreciate it.
> > >
> > > > **Q3**. The authors claim that there is no explicit ground truth as the target, but the final answer is not necessarily the target either. Given that preference optimization relies on rewards for evaluation, generation with high reward is the target. The final answer does not necessarily enjoy high reward.
> > >
> > > **A3**. We do not claim that the final answers are the true targets. **We use the final responses as a proxy to probe how different objectives move the model in a direction that improves the likelihood of what the model eventually produces**. Moreover, the true PO target remains high‑reward behavior (as defined by the reward model). Our gradient alignment condition is intended to capture the stability of the optimization dynamics toward these targets, not to redefine the targets themselves. We view this as a promising way to analyze learning behaviors, and we would be happy to consider and discuss alternative strategies if you have other suggestions.
> > >
> > > > **Q4**. I would like to clarify the theory foundation of DPO. When
> > > $\pi$ is not optimal, the induced reward $r$ is indeed not the true latent reward $r^*$, but the policy $\pi$ is optimal under its induced reward $r$. Given that any policy is optimal with respect to its induced reward, it only requires to perform reward modeling for the induced reward. When the induced reward is (nearly) the true latent reward, the policy is (nearly) optimal (under the true latent reward).
> > >
> > > **A4**. We are grateful for, and genuinely impressed by, your generous advice. We will keep it in mind and be more careful on this point in future related discussions. Thanks again for your generous suggestions and comments, which mean a lot to us and have helped improve the quality of our paper.

---

> ### Author Response · Authors · 2025-11-24
>
> Dear Reviewer CKPZ,
>
> We sincerely thank you for your efforts in reviewing our work! Please let us know if you need any further information or if there are additional points you would like to discuss with us.
>
> Thank you very much!
>
> Best regards,
>
> Authors of #4235

---

> ### Comment · Reviewer_CKPZ · 2025-11-26
>
> > A3. Using the final answer rather than the ground truth is one of our key design choices in our analytical framework. We are so glad that you raised this queston and would like to explain it further as follows.
>
> The authors claim that there is no explicit ground truth as the target, but the final answer is not necessarily the target either.
> Given that preference optimization relies on rewards for evaluation, generation with high reward is the target.
> The final answer does not necessarily enjoy high reward.

---

### Official Review · Reviewer_pfMR · 2025-10-23

**Soundness:** 3
**Presentation:** 3
**Contribution:** 3
**Rating:** 6
**Confidence:** 4

**Summary:**

This paper analyzes the optimization dynamics of Direct Preference Optimization (DPO) and Proximal Policy Optimization (PPO) to elucidate the distinct roles of positive learning, negative learning, and loss reweighting. The authors introduce a "gradient alignment" metric to investigate how learning targets evolve during training. The analysis reveals that in DPO, positive/negative learning jointly shape targets while loss reweighting acts as a regularizer. In contrast, PPO uses negative learning to aid exploration, and its loss reweighting differentiates the roles of token groups in updating the policy. The authors substantiate these findings with ablation studies examining the practical performance implications of controlling these dynamic components.

**Strengths:**

1. The paper provides a deep, mechanistic explanation for the oft-discussed differences between DPO and PPO by skillfully analyzing their respective training dynamics.
2. The introduction of the 'gradient alignment' metric is a notable contribution, offering an effective method to quantify and inspect the optimization dynamics of preference alignment algorithms.
3. The findings are clear and insightful, providing actionable explanations for the distinct roles of positive learning, negative learning, and loss reweighting.
4. The paper's analytical claims are well-supported by sufficient empirical validation, including targeted ablation studies that connect the observed dynamics to practical performance.

**Weaknesses:**

1. The paper provides extensive empirical analysis, but it lacks a rigorous theoretical foundation to formally explain the underlying reasons for the observed phenomena.
2. The analysis could be strengthened by incorporating the distribution of key data properties. For instance, analyzing the distributions of the DPO reweighting term ($\omega$) and the PPO absolute advantage ($|\hat A|$), both globally and within subgroups, would provide a more complete picture of their impact.
3. The 'gradient alignment' metric is a first-order approximation that does not account for the adaptive, non-linear dynamics of optimizers like AdamW or the non-convex landscape.
4. Minor Issues on Presentation:
* The conclusion (Section 5) is somewhat lengthy and could be compressed. This would create space to either expand the main analysis or move valuable insights from the appendices (e.g., parts of Appendix D) into the main paper.
* The experimental cases (e.g., "Case 1-3") in Figure 4 are not clearly explained in the text, making the results difficult to interpret fully.

**Questions:**

1. To strengthen the analysis on reweighting, could the authors show the distributions of the DPO term ($\omega$) and the PPO absolute advantage ($|\hat A|$)? It would be insightful to see this for the entire dataset and within the 'top', 'middle', and 'bottom' subgroups. This might also help justify the current split into three equal-sized groups, or perhaps suggest a more natural, data-driven way to segment the data.
2. The cDPO (controlled DPO) in Appendix E explores a gradual shift from positive to negative learning. As a clearer ablation to test the "role-switching" hypothesis, what would be the result of a hard switch? (e.g., training only with the positive learning component for the first half of training, and only with the negative component for the second half).
3. In lines 349-351, the authors state that in PPO, 'positive learning is stable in shaping the learning targets.' This is a key finding. Could the authors provide any further mathematical derivation or theoretical intuition to explain why this is the case, while negative learning's role is relegated to exploration?

---

> ### Author Response · Authors · 2025-11-21
>
> We sincerely thank you for your constructive comments and generous support, which mean a great deal to us. Our responses are provided below, and we hope they can sufficiently address your concerns.
>
> > **Q1**. The paper provides extensive empirical analysis, but it lacks a rigorous theoretical foundation to formally explain the underlying reasons for the observed phenomena.
>
> **A1**. We position our paper as an analysis and interpretation study rather than a theoretical work. Accordingly, **our primary goal in the theoretical aspect is to identify a suitable analyzing tool**, as in Section 2.1. It can be further generalized to capture the accmulated effects of gradient updates, as detailed in A3 of our response to Reviewer wNW2. Our responses to Q5 below also provide some insights into the gradient flow associated with positive and negative learning, respectively.
>
> For the theoretical foundation of learning dynamics, although it is undoubtedly an important aspect, it truly falls beyond the scope of this work. To the best of our knowledge, **the literature remains the lack of a proper derivation framework capable of characterizing mutually impacted dynamics**, where different components interact and influence one another during training. As our study takes a new perspective over previous PO studies, i.e., studying the learning dynamics in achieving final responses, traditional convergence-based or stability-based analyses are not that suitable for this purpose. While we sincerely appreciate and understand your interest in further theoretical justification, sadly, we need to take this as an important direction for future research.
>
>
> > **Q2**. To strengthen the analysis on reweighting, could the authors show the distributions of the DPO term ($\omega$) and the PPO absolute advantage ($\vert\hat A\vert$)? It would be insightful to see this for the entire dataset and within the 'top', 'middle', and 'bottom' subgroups. This might also help justify the current split into three equal-sized groups, or perhaps suggest a more natural, data-driven way to segment the data.
>
> **A2**. We report the mean weight values for $\omega$ and $\vert\hat A\vert$ as follows. As observed, both $\omega$ and $\vert\hat A\vert$ decrease as training progresses, but the scales of DPO is much larger than those for PPO. However, since the data distributions (especially the responses) are inherently different, it is difficult to compare DPO and PPO on a per-data point manner. Nonetheless, we truly believe your assumption is reasonable and promising, and we will work for a better way in comparing the weighting strategies of DPO and PPO in future work.
>
> | Step | $\omega$ |
> |------|-----------------|
> | 1000 | 0.358613 |
> | 2000 | 0.298699 |
> | 3000 | 0.254373 |
> | 4000 | 0.217262 |
> | 5000 | 0.202154 |
> | 6000 | 0.188888 |
> | 7000 | 0.183865 |
>
>
> | Step | $\vert\hat A\vert$ |
> |------|------------------|
> | 400  | 0.022356 |
> | 800  |  0.022101 |
> | 1200 | 0.020993 |
> | 1600 | 0.019962 |
> | 2000 | 0.018519 |
> | 2400 |  0.018256 |
>
>
> > **Q3**. The 'gradient alignment' metric is a first-order approximation that does not account for the adaptive, non-linear dynamics of optimizers like AdamW or the non-convex landscape.
>
> **A3**. We completely agree that the first-order assumption is a limitation when using Adam-based optimizers, and we have acknowledged this factor in Appendix A. **A more grounded alternative would be the Fisher Kernel**, which incoorporates the Fisher information matrix to measure the underlying geometry of the parameter space. In fact, at the early stage of this study, we have considered the use of it, but its computational cost is extremely high for LLMs. We also considered the possibility of applying the Fisher Kernel only to the top layers of the model. Nevertheless, as shown in Figure 9(b), learning behaviors differ largely across layers, and restrciting the analysis to the top layers could introduce additional sources of bias. Therefore, **we opted for a practical trade‑off between theoretical rigor and computational feasibility**, using the gradient alignment condition as our final solution.
>
> On the other side, **we believe that, though limited by the first-order assumption, the gradient alignment condition serves as an interpretable tool**. The insights drawn from this condition directly inspired the development of cDPO, cPPO, and hPPO, where we have demonstrated their potential for acheieving futher improvements in performance. However, we acknowledge that a more rigorous tool incorporating optimizer dynamics and curvature effects would further enrich our understanding. We plan to explore these extensions in our future research.

---

> ### Author Response · Authors · 2025-11-21
>
> > **Q4**. The cDPO (controlled DPO) in Appendix E explores a gradual shift from positive to negative learning. As a clearer ablation to test the "role-switching" hypothesis, what would be the result of a hard switch? (e.g., training only with the positive learning component for the first half of training, and only with the negative component for the second half).
>
> **A4**. Many thanks for your question! Fortunately, in Figure 10 (h), we reported a case of a hard switch that may match your interest. As observed, **a sudden transition from positive to negative learning leads to rapid performance deterioration**, falling far below that before PO training. In contrast, when the transition is conducted smoothly, the performance showed the potential for notable improvement. We conjecture that **a smooth control enables the model to somehow preserve the beneficial behaviors achieved during positive learning, thereby reaching a Pareto-like solution**. In our revision, we will further highlight this hard‑switch example and provide additional discussions on its difference from the soft‑switch counterparts.
>
> > **Q5**. In lines 349-351, the authors state that in PPO, 'positive learning is stable in shaping the learning targets.' This is a key finding. Could the authors provide any further mathematical derivation or theoretical intuition to explain why this is the case, while negative learning's role is relegated to exploration?
>
> **A5**. We can verify that positive learning shapes the learning targets, whereas the behavior of negative learning is more subtle. Below, we outline a reasonable analysis framework consisting of two parts.
>
> **Target Shaping**. We consider the model $\pi(y|x;\theta)$ parameterized the conditional probability distribution of $y$ given $x$, along with the empirical data distribution $p(x,y)$ used for model training. For positive learning, when analyzing the distribution evolution of $\pi$ over $t$, we have the gradient flow of $\frac{d\theta}{dt}=\eta\mathbb E_{p(x,y)}\nabla\log\pi(y|x;\theta)$. Based on the chain rule, we further have $\frac{d\pi(y|x;\theta)}{dt}=\eta\pi(y|x;\theta)\mathbb E_{p(x',y')}[\nabla\log\pi(y'|x';\theta)^\top\nabla\log\pi(y|x;\theta)]$. Assuming a diagonal Fisher matrix, we approximate $\mathbb E_{p(x',y')}[\nabla\log\pi(y'|x';\theta)^\top\nabla\log\pi(y|x;\theta)]\approx c(x)[\frac{p(y|x)}{\pi(y|x;\theta)}-1]$, where $c(x)$ is some positive scalar function. Thus, **we have $\frac{d\pi(y|x;\theta)}{dt}=\eta c(x)[p(y|x)-\pi(y|x;\theta)]$ for positive learning, where $\pi(y|x;\theta)$ moves toward $p(y|x)$ as the target**. Similarly, **we have $\frac{d\pi(y|x;\theta)}{dt}=-\eta c(x)[p(y|x)-\pi(y|x;\theta)]$ for negative learning, in which case $\pi(y|x;\theta)$ deviates from $p(y|x)$**.
>
> **Learning Stability**. Upon stability, we conduct the conventional small-perturbation analysis. For positive learning, we add a small perturbation $\epsilon(y|x)$ to the optimal $\pi(y|x; \theta^\*)$ and then linearize the equation in $\epsilon$. Since $\frac{d\pi(y|x;\theta^\*)}{dt}=0$, we have $\pi(y|x;\theta^*)=p(y|x)$. Adding $\epsilon$ to $\frac{d\pi(y|x;\theta)}{dt}$, we have $\frac{d[\pi(y|x;\theta)+\epsilon(y|x)]}{dt}=-\eta c(x)\epsilon(y|x)$ and thus $\dot \epsilon(y|x)=-\eta c(x)\epsilon(y|x)$. Solving this linear DOE function leads to $\epsilon(t)=\epsilon(0)e^{-\eta c(x)t}$. Similarly, we have $\epsilon(t)=\epsilon(0)e^{\eta c(x)t}$ for negative learning. Since $c(x)>0$ and $\eta>0$, **we have $\epsilon(0)e^{-\eta c(x)t}\rightarrow0$ as $t\rightarrow\infty$, indicating stability for positive learning**. In contrast, **$\epsilon(0)e^{\eta c(x)t}\rightarrow\infty$ as $t\rightarrow\infty$, indicating instability for negative learning**.
>
> We acknowledge that the above derivations are indeed straightforward and simple. However, since the primary focus of this paper lies in empirical analysis rather than theoretical exploration, we leave more rigorous derivations and the development of a comprehensive theoretical framework for future work.
>
> > **Q6**. Minor Issues on Presentation.
>
> **A6**. We apologize for the confusion and deeply appreciate your suggestions regarding the organization of this paper. In Figure 4, **each case corresponds to the win‑rate curve under a specific hyperparameter configuration**, with detailed and extended results provided in Appendix E in our paper. We agree that referring to these configurations simply as “Case 1–3” may cause misunderstanding, and we will revise the presentation to clearly indicate the associated hyperparameter settings. We sincerely appreciate your detailed feedback and the time you spent reviewing our work, which means a great deal to us.

---

> ### Author Response · Authors · 2025-11-25
>
> Dear Reviewer pfMR,
>
> We sincerely thank you for your efforts in reviewing our work and for your great support! Please let us know if you need any further information or if there are additional points you would like to discuss with us. We would be glad to answer any questions of your interest.
>
> Thank you very much!
>
> Best regards,
>
> Authors of #4235

---

> > ### Comment · Reviewer_pfMR · 2025-11-25
> >
> > Thanks for the detailed response. I acknowledge that addressing the theoretical and empirical aspects of Q2 and Q3 within the short rebuttal timeframe is challenging. Regarding Q3, I suggest the authors consult [A] for insights on the Fisher kernel or Hessian matrix approximation. Also, it would be better to incorporate the clarifications provided in A5 and A6 into the revision. I will maintain my positive score.
> >
> > [A] Zhang, Yushun, et al. "Why transformers need Adam: A Hessian perspective." Advances in neural information processing systems 37 (2024)

---

> > > ### Author Response · Authors · 2025-11-27
> > >
> > > Sincere thanks for your feedbacks and generous support. We will cite [A] and add a brief discussion highlighting Fisher kernel as an important direction for future research. We have also updated our manuscript: A5 is now included in Appendix E, and A6 is incorporated into Figure 4 and Section 5.

---

### Official Review · Reviewer_wNW2 · 2025-11-01

**Soundness:** 3
**Presentation:** 4
**Contribution:** 2
**Rating:** 2
**Confidence:** 3

**Summary:**

The paper studies what preference optimization (PO) does by analyzing optimization dynamics for DPO and PPO through a gradient-alignment metric that measures the dot product between the PO objective gradient and the gradient of expected NLL on final responses. It reports that DPO behaves like supervised learning with targets implicitly shaped by both positive and negative learning, while PPO behaves like reinforcement learning with exploration near orthogonal targets; loss reweighting acts more like regularization in DPO and carries token-level information in PPO. The authors further test behavior-control variants (cDPO, cPPO, hPPO) and show illustrative win-rate gains on AlpacaEval with Pythia-2.8B.

**Strengths:**

- Well-structured decomposition of PO into positive/negative learning and reweights; which connects intuitively to training heuristics.
- The *gradient alignment* tool is simple, and allows concrete insights.
- Evaluates variants of PO (cDPO, cPPO, hPPO) from the insights acquired from the analysis.

**Weaknesses:**

- Insufficient breadth and scale of experiments
    - The paper uses a single base model (Pythia-2.8b) and narrow task sets. The claims in the paper about "what PO is doing" should be tested on larger models, multiple families, and varied domains.
    - Although the motivation of the paper seems promising, empirical proof of PO tendency should be backed up with much more depth.
- Lack of theoretical framing
    - Tightening the theoretical relation between $G$ and the performance can strenghthen the motivation of the paper, when extensive experiments is infeasible.

**Questions:**

- Refer to the weakness section.

---

> ### Author Response · Authors · 2025-11-21
>
> Sincere thanks for your constructive comments, and we hope the following feedbacks can address your concerns.
>
>
> > **Q1**. The paper uses a single base model (Pythia-2.8B) and narrow task sets. The claims in the paper about "what PO is doing" should be tested on larger models, multiple families, and varied domains.
>
> **A1**. We used Pythia as the base model because it is open-sourced with detailed documentation of its training and fine-tuning procedures, making it well-suited for rigorous verification. For instance, we know that this base model has not been trained with any preference optimization procedures, thereby eliminating potential confounding factors. Few base models are this transparent, which is why we only adopted Pythia in our experiments.
>
> On the other hand, we fully agree that conducting experiments on a broader range of models would further strengthen the generality of our conclusions. Therefore, following your kind suggestion, **we additionally tested the gradient alignment conditions on Qwen3-1.7B**, with the results summarized as follows.
>
> **DPO — Gradient Alignment**
>
> | Training Step | NEG (Negative) | POS (Positive) |TOT (Total)|
> |------|--------|----------------|----------------|
> | 30 | -1.948e+04| 2.848e+04|9.038e+03|
> | 60 | -4.267e+03| 1.032e+04|6.056e+03|
> | 90 | -4.430e+02| 5.345e+03|4.895e+03|
> | 120| -6.467e+01| 6.434e+03|6.368e+03|
> | 150| 7.253e+03| -5.907e+03|1.344e+03|
> | 180| 6.096e+03| -4.036e+03|2.063e+03|
> | 210| 6.372e+03| -5.467e+03|9.086e+02|
> | 240| 6.426e+03| -5.435e+03|1.002e+03|
>
>
> | Training Step | TOP | MID |BOT|
> |----|----------|-----|---------|
> | 30 | 6.265e+03| 2.388e+03|3.745e+02|
> | 60 |3.233e+03| 2.250e+03|5.705e+02|
> | 90 | 2.672e+03| 1.847e+03|3.759e+02|
> | 120| 4.647e+03| 1.361e+03|3.580e+02|
> | 150| 2.201e+01| 1.045e+03|3.185e+02|
> | 180| 3.025e+02| 1.471e+03|2.862e+02|
> | 210| -4.842e+02|1.098e+03|2.915e+02|
> | 240| -3.968e+02|9.858e+02|3.385e+02|
>
>
> **PPO — Gradient Alignment**
>
> | Training Step | NEG (Negative) | POS (Positive) |TOT (Total)|
> |------|--------|----------------|----------------|
> | 40| -7.303e+00| 6.007e+00|-1.298e+00|
> | 80| -7.589e+00| 5.937e+00|-1.651e+00|
> |120| -7.925e+00| 6.783e+00|-1.140e+00|
> |160| -8.330e+00| 6.555e+00|-1.772e+00|
> |200| -8.500e+00| 6.617e+00|-1.882e+00|
> |240| -8.595e+00| 6.717e+00|-1.877e+00|
> |280| -8.491e+00| 6.919e+00|-1.569e+00|
> |320| -8.625e+00| 7.083e+00|-1.543e+00|
> |360| -8.247e+00| 6.974e+00|-1.268e+00|
> |400| -8.227e+00| 6.648e+00|-1.574e+00|
>
> | Training Step | TOP | MID |BOT|
> |---------------|-----|-----|---|
> | 40 | -1.234e+00|-2.800e-01|2.026e-02|
> | 80 | -8.211e-01| -2.002e-01|4.441e-03|
> | 120| -5.304e-01| -3.095e-01|5.317e-03|
> | 160| -7.414e-01| -2.437e-01|-2.712e-03|
> | 200| -1.033e+00| -2.669e-01|-4.364e-02|
> | 240| -3.575e-01| -1.339e-01|-5.020e-02|
> | 280| -2.365e-01|-2.141e-02|2.325e-03|
> | 320| -2.321e-01|-2.975e-01|-6.778e-02|
> | 360| -9.123e-01|-1.090e-01|-2.179e-02|
> | 400| -2.363e-01|-6.816e-02|-2.841e-02
>
>
> As observed, **the conclusions drawn from Pythia also explain the learning dynamics in Qwen, demonstrating that our findings are general and reliable**.
>
> Furthermore, **we also evaluate cPPO, cDPO, and hPPO on Qwen3-1.7B and Llama-3-8B**, comparing their win rates against the original PPO and DPO under their best-performing settings, as summarized below.
>
>
> **Pythia-2.8B**
> |PO Method|Win Rate (%) ↑ (mean ± std)|
> |------|--------------|
> | DPO  |76.71 ± 0.20|
> | cDPO |**79.88** ± 0.13|
> | PPO  |55.96 ± 0.24|
> | cPPO |**57.76** ± 0.10|
> | hPPO |56.89 ± 0.14|
>
> **Qwen3-1.7B**
> | PO Method | Win Rate (%) ↑ (mean ± std) |
> |-----|----|
> | DPO | 72.14 ± 0.16|
> | cDPO| **87.91** ± 0.20|
> | PPO | 57.10 ± 0.18|
> | cPPO| **61.77** ± 0.15|
> | hPPO| 61.17 ± 0.12|
>
>
> **Llama-3-8B**
> |PO Method|Win Rate (%) ↑ (mean ± std))|
> |------|----|
> | DPO  | 67.71 ± 0.27|
> | cDPO | **69.54** ± 0.06|
> | PPO  | 55.60 ± 0.34|
> | cPPO | **57.04** ± 0.07|
> | hPPO | 56.99 ± 0.19|
>
> We will include the performance and alignment-condition curves in our revised version to provide clearer illustrations. We will explore additional models and domains in future work.
>
> > **Q2**. Although the motivation of the paper seems promising, empirical proof of PO tendency should be backed up with much more depth.
>
> **A2**. The scope of our paper **has covered not only the objective aspect but also the data aspect (e.g., Figures 1c,  2c, and 4d-4f) and the model aspect (e.g., Figure 9)**. We hope that the addtional results further provided in A1 could further strengthen our analysis. We also welcome any further suggestions regarding additional experiments and analyses the reviewers would like to see, and we would be glad to discuss ways to broaden the scope and depth of our paper.

---

> ### Author Response · Authors · 2025-11-21
>
> > **Q3**. Tightening the theoretical relation between $G$ and the performance can strenghthen the motivation of the paper, when extensive experiments is infeasible.
>
> **A3**. In Section 2.1, we showed that for each update step, the gradient alignment condition could indicate performance changes under a first-order approximation. **This serves as a motivating derivation and can be further formalized to examine the cumulative effects of gradients**, which aligns more closely with our discussion in Section 3. An outline of our refined derivation is provided as follows.
>
> When testing the parameter changes after two consecutive PO steps, we have $\theta_{t+2}=\theta_{t}-\eta\nabla\mathcal L(\mathbf\theta_{t})-\eta\nabla\mathcal L(\mathbf\theta_{t}-\eta\nabla\mathcal L(\mathbf\theta_{t}))$, which can be further approximated as $\theta_{t+2}\approx\theta_{t}-\eta(2\nabla\mathcal L(\mathbf\theta_{t})-\eta\nabla^2\mathcal L(\mathbf\theta_{t})\nabla\mathcal L(\mathbf\theta_{t}))$. This formulation can be generalized for the accumulations of $T$ steps, which is $\theta_{T}\approx\theta_{0}-\eta T\nabla\mathcal L(\mathbf\theta_{0})+\sum_t\phi_{t}$, where $\phi_{t}$ is defined recursively as $\phi_{t}=-\eta\nabla^2\mathcal L(\theta_0)(-\eta\sum_t\nabla\mathcal L(\mathcal D;\theta_0)+\sum_{t'}\phi_{t'})$ with $\phi_{0}=0$. When $\eta$ is small, we have $\phi_{t}\approx\eta^2T\nabla^2\mathcal L(\theta_0)\nabla\mathcal L(\theta_0)$.
>
> Thus, we have $\theta_T\approx\theta_0-\eta AT\mathcal L(\theta_0)$, where $A=I-\eta T \nabla^2\mathcal L(\theta_0)$ and $I$ is the identify matrix. Regarding the performance measured by the negative log-likelihood of final responses, we have $\mathcal R(\theta_{T})-\mathcal R(\theta_0)\approx-\eta T\nabla R(\theta_0)^\top A\nabla\mathcal L(\theta_0)$. The RHS is lower and upper bounded by $\lambda_1\vert\vert\nabla R(\theta_0)\vert\vert\cdot\vert\vert\nabla\mathcal L(\theta_{0})\vert\vert$ and $\lambda_2\vert\vert\nabla R(\theta_{0})\vert\vert\cdot\vert\vert\nabla\mathcal L(\theta_{0})\vert\vert$, where $\lambda_1$ and $\lambda_2$ are the minimal and maximal eigenvalues of $A$.
>
> One step further, with small $\eta$, there exists a $\lambda\in[\lambda_1,\lambda_2]$ such that $-\eta T\lambda\nabla R(\theta_0)^\top\nabla\mathcal L(\theta_0)$ is a good approximation of $-\eta T\nabla R(\theta_0)^\top A\nabla\mathcal L(\theta_0)$. Overall, if the learning rate $\eta$ is sufficiently small, the performance change under the accmulated gradients is characterized by $-\eta \lambda T \nabla R(\theta_0)^\top\nabla\mathcal L(\theta_0)$. Since $\eta \lambda T$ is a constant, **we have the same gradient alignment condition as Eq. 3 in our paper**. The above analysis is motivated by [1].
>
> [1] Thudi et al. Unrolling SGD: Understanding factors influencing machine unlearning. In EuroS&P, 2022.
>
>
> At last, we sincerely appreciate your thoughtfulness in considering that we may have limited computational resources for additional experiments. We have provided additional results in A1, and we will explore more base models in the future. We also look forward to any further discussions and suggestions, which will be invaluable for improving the quality of this paper.

---

> ### Author Response · Authors · 2025-11-24
>
> Dear Reviewer wNW2,
>
> We sincerely thank you for your efforts in reviewing our work! Please let us know if you need any further information or if there are additional points you would like to discuss with us.
>
> Thank you very much!
>
> Best regards,
>
> Authors of #4235

---

> ### Author Response · Authors · 2025-11-27
>
> Dear Reviewer wNW2,
>
> We have added additional results for more models (Qwen3-1.7B and Llama-3-8B) in Appendices C and D, and included further theoretical analyses (gradient alignment and component behaviors) in Appendix E, which we believe address your main concerns.
>
> As the rebuttal period is ending soon (Dec 04 06:00 AM), we would like to confirm whether our responses have resolved your concerns and whether you have any remaining questions you would like us to clarify. We would be very happy to continue the discussion.
>
> Thank you for your time and effort in reviewing our work, and we look forward to your reply!
>
> Best regards,
>
> Authors of #4235

---

> > ### Comment · Reviewer_wNW2 · 2025-11-28
> >
> > I thank the authors for the detailed rebuttal and the additional experiments.
> >
> > ### 1. Gradient‐alignment remains a surrogate metric.
> > While gradient alignment provides an intuitive and useful lens on PO dynamics, it is nevertheless an indirect proxy for performance. The theoretical extension in the rebuttal still relies on first-order linearization assumptions. Therefore, strong empirical breadth is needed to substantiate claims about “what PO is doing,” beyond confirming that the surrogate behaves consistently across a few models.
> >
> > ### 2. Breadth across tasks and domains is still limited.
> > I appreciate the addition of Qwen3-1.7B and Llama-3-8B. However, these additions remain small in scope relative to the ambition of explaining general PO dynamics. Demonstrating robustness across a larger variety of tasks, domains, and preference datasets would greatly strengthen the claims. Currently, the evidence base remains narrow.
> >
> > Even with the additional experiments, I do not think it is safe to take the paper’s conclusions as granted. The empirical foundation is still too narrow to support broad claims about the nature of preference optimization across model families, scales, and domains. The insights are suggestive, but not yet definitive.

---

### Meta-Review · Area_Chair_Ekhb · 2026-01-07

**Summary:**

The main decision-critical concerns:
- limited experimental breadth/scale: initially largely one backbone and narrow tasks; even after rebuttal, still not enough to justify general conclusions. (wNW2)
- surrogate limitations: gradient alignment is a surrogate/first-order proxy whose link to performance is not theoretically tight, and thus requires stronger empirical substantiation. (wNW2 & pfMR)
- less convinced claims/theoretical framing: insufficiently rigorous theoretical framing for several claims and design choices (notably in how SFT vs RL is characterized, and why final-answer log-prob is the target of analysis). (CKPZ, pfMR）

**Reviewer Concerns:**

Concerns that were addressed (partially) by the rebuttal:
- Added experiments / broader backbones: additional model families/sizes (e.g., Qwen and Llama variants), which helped some reviewers feel the qualitative trends were consistent (i8HD acknowledges consistency; wNW2 appreciates the additions but still finds scope limited).
- Statistical reporting: multi-run variability (std dev) for win-rate experiments, alleviating part of the “are these differences just noise?” concern (i8HD explicitly acknowledges this).

Still outstanding:
- insufficient breadth: Even with added backbones, the task/domain/preference-dataset coverage is too narrow to claim general PO dynamics across scales/families/domains. (wNW2)
- surrogate metric limitations: gradient alignment remains a proxy with first-order/linearization assumptions; reviewers want either tighter theory or much broader empirical confirmation that the proxy meaningfully predicts or explains performance. (wNW2 & pfMR).
-  less convinced claims/theoretical framing: there are some logical and conceptual gaps (e.g., the SFT-vs-RL distinction framed via “stable targets,” differentiability discussion, justification for using final-answer log-prob, and theory claims around DPO, distributions of DPO reweighting term and PPO advantages, subgroup stratification rationale). (CKPZ & pfMR)

**Reviewer Scores:**

- wNW2 (original score 2): No change, as the reviewer explicitly states the evidence is still too narrow and conclusions are not safe to take as granted.
- pfMR (original score 6): No change, as the reviewer requests revision-level improvements (theory/distributional analyses/clarity).
- CKPZ (original score 2): No change, as the reviewer states the Key logical/theoretical objections remain even after rebuttal.
- i8HD (original score 6): No change

---

### Decision · Program_Chairs · 2026-01-26

Reject